# Determination of enhancement ratios of HCOOH relative to CO in biomass burning plumes by the Infrared Atmospheric Sounding Interferometer (IASI)

5  Matthieu Pommier[1,*], Cathy Clerbaux[1,2], Pierre-Francois Coheur[2]

1 LATMOS/IPSL, UPMC Univ. Paris 06 Sorbonne Universités, UVSQ, CNRS, Paris, France
2 Spectroscopie de l'Atmosphère, Chimie Quantique et Photophysique, Université Libre de Bruxelles (ULB), Brussels, Belgium
10  *now at: Norwegian Meteorological Institute, Oslo, Norway

*Correspondence to:* Matthieu Pommier (matthieup@met.no)

15  **Abstract.** Formic acid (HCOOH) concentrations are often underestimated by models and its chemistry is highly uncertain. HCOOH is, however, among the most abundant atmospheric volatile organic compounds and it is potentially responsible for rain acidity in remote areas. HCOOH data from the Infrared Atmospheric Sounding Interferometer (IASI) are analyzed from 2008 to 2014, to estimate enhancement ratios from biomass burning emissions over seven regions. Fire-affected HCOOH and CO total columns are defined by combining total columns from IASI, geographic location of the fires from MODerate resolution Imaging Spectroradiometer (MODIS), and the surface wind speed field from the European Centre for Medium-Range Weather Forecasts (ECMWF). Robust correlations are found between these fire-affected HCOOH and CO total columns over the selected biomass burning regions, allowing the calculation of enhancement ratios equal to $7.30\times10^{-3}\pm0.08\times10^{-3}$ mol/mol over Amazonia, $11.10\times10^{-3}\pm1.37\times10^{-3}$ mol/mol over Australia, $6.80\times10^{-3}\pm0.44\times10^{-3}$ mol/mol over India, $5.80\times10^{-3}\pm0.15\times10^{-3}$ mol/mol over Southern East Asia, $4.00\times10^{-3}\pm0.19\times10^{-3}$ mol/mol over Northern Africa, $5.00\times10^{-3}\pm0.13\times10^{-3}$ mol/mol over Southern Africa, and $4.40\times10^{-3}\pm0.09\times10^{-3}$ mol/mol over Siberia, in a fair agreement with previous studies. In comparison with referenced emission ratios, it is also shown that the selected agricultural burning plumes captured by IASI over India and Southern East Asia correspond to recent plumes where the chemistry or the sink does not occur. An additional classification of the enhancement ratios by type of fuel burned is also provided, showing a diverse origin of the plumes sampled by IASI, especially over Amazonia and Siberia. The variability in the enhancement ratios by biome over the different regions show that the levels of HCOOH and CO do not only depend on the fuel types.

## 1. Introduction

Formic acid (HCOOH) is one of the most abundant carboxylic acids present in the atmosphere. HCOOH is mainly removed from the troposphere through wet and dry deposition, and to a lesser extent by the OH radical. It is a relatively short-lived species with an average lifetime in the troposphere of 3–4 days (Paulot et al., 2011; Stavrakou et al., 2012). HCOOH contributes a large fraction to acidity in precipitation in remote areas (e.g. Andreae et al., 1988).

HCOOH is mainly a secondary product from other organic precursors. The largest global source of HCOOH is biogenic and follows the emissions of isoprene, monoterpenes, other terminal alkenes (e.g., Neeb et al., 1997; Lee et al., 2006; Paulot et al., 2011), alkynes (Hatakeyama et al., 1986; Bohn et al., 1996), and acetaldehyde (Andrews et al., 2012; Clubb et al., 2012). There are also small direct emissions by vegetation (Keene and Galloway, 1984, 1988; Gabriel et al., 1999) and biomass burning (e.g. Goode et al., 2000). Other studies highlighted the existence of other sources, such as from ants (Graedel and Eisner, 1988), dry savanna soils (Sanhueza and Andreae, 1991), motor vehicles (Kawamura et al., 1985; Grosjean, 1989), abiological formation on rock surfaces (Ohta et al., 2000) and cloud processing (Chameides et al., 1983). Their contributions are very uncertain and most are probably minor.

More generally there are still large uncertainties on the sources and sinks of HCOOH, and on the relative contribution of anthropogenic and natural sources, despite the fact that recent progress has been made possible by using the synergy between atmospheric models and satellite data (e.g. Stavrakou et al., 2012; Chaliyakunnel et al., 2016). These uncertainties have an impact on our understanding of the HCOOH tropospheric chemistry, as on the oxidizing capacity of the atmosphere (i.e. the chemistry of OH in cloud water - Jacob, 1986; the heterogeneous oxidation of organic aerosols - Paulot et al., 2011) or the origin of the acid rains. One of the large uncertainties in the HCOOH tropospheric budget seems to be the underestimation of the emissions from forest fires, as recently suggested by Stavrakou et al. (2012), Cady-Pereira et al. (2014) and Chaliyakunnel et al (2016).

One way to estimate the atmospheric emissions of pyrogenic species is the use of emission factors. The emission factors are often obtained from ground and airborne measurements or from small fires burned under controlled laboratory conditions. The emission factors can also be derived from enhancement ratios of the target species relative to a reference species, which is often carbon monoxide (CO) or carbon dioxide ($CO_2$) due to their long lifetime (e.g. Hurst et al., 1994) and are based on the characteristic of the combustible and hence depend on the type of biomass burning. However, the difference between an emission ratio and an enhancement ratio is that emission ratios are calculated from measurements at the time of emission and enhancement ratios are related to the ongoing chemistry. To correctly convert these enhancement ratios to emission ratios, the decay of the chemical species need to be taken into account or assumptions need to be made, suggesting that the enhancement ratios are equivalent to emission ratios, hence measured at the source and not impacted by chemistry.

Compilations of numerous enhancements ratios, emission ratios and emission factors for several trace gases from measurements at various locations world-wide are published regularly (e.g. Akagi et al., 2011) in order to facilitate their use in Chemistry Transport Models.

There has been a recent interest in calculating enhancement ratios and emission factors from satellite data (e.g. Rinsland et al., 2007; Coheur et al., 2009; Tereszchuk et al., 2011). The above difficulty of inferring emission factors using the satellite observations comes from the fact that these observations are indeed typically made in the free/upper troposphere and further downwind of the fires. The fact that satellite mainly probe transported plumes where chemistry modifies the original composition explains why the use of enhancement ratio is more relevant than emission ratio.

Only a few papers have reported on the use of satellite retrievals to study tropospheric HCOOH including the nadir-viewing Infrared Atmospheric Sounding Interferometer (IASI) (e.g. Razavi et al., 2011; Stavrakou et al., 2012; R'Honi et al., 2013; Pommier et al., 2016), and Tropospheric Emission Spectrometer (TES) (e.g. Cady-Pereira et al., 2014; Chaliyakunnel et al., 2016). Other studies have used the solar occultation Atmospheric Chemistry Experiment – Fourier Transform Spectrometer (ACE-FTS) which measures the atmospheric composition in the upper troposphere (e.g. Rinsland et al., 2006; Gonzalez Abad et al., 2009; Tereszchuk et al., 2011; 2013) and the Michelson Interferometer for Passive Atmospheric Sounding (MIPAS) limb instrument, which is sensitive to around 10 km (Grutter et al., 2010).

These Infrared (IR) sounders have limited vertical sensitivity as compared to ground-based or airborne measurements but their spatial coverage represents a major advantage, which allows observation of remote regions which are sparsely studied by field measurements, like the biomass burning regions.

This work aims to provide a list of enhancement ratios of HCOOH relative to CO over several biomass burning regions. For this, we analyzed seven years of IASI measurements, between 2008 and 2014. Section 2 describes the IASI satellite mission and the retrieval characteristics for the CO and the HCOOH total columns. Section 3 presents the fire product used from the MODerate resolution Imaging Spectroradiometer (MODIS) to identify the fire locations. Section 4 details the methodology used to identify of the IASI fire-affected observations. In Section 5 we describe and analyze the enhancements ratios obtained from the IASI measurements, including an analysis of these ratios by type of fuel burned and we compare these values to those available in the literature. Finally, the conclusions are presented in Section 6.

## 2. HCOOH and CO columns from IASI

### 2.1 The IASI mission

IASI is a nadir-viewing Fourier Transform Spectrometer. Two models are currently in orbit. The first model (IASI-A), was launched onboard the METOP-A platform in October 2006. The second instrument was launched in September 2012 onboard METOP-B. Owing to its wide swath, IASI delivers near global coverage twice a day with observation at around 09:30 and 21:30 local time. Each atmospheric view is composed of 2×2 circular pixels with a 12 km footprint diameter, spaced out by 50 km at nadir. IASI measures in the thermal infrared part of the spectrum, between 645 and 2760 cm$^{-1}$. It records radiance from the Earth's surface and the atmosphere with an apodized spectral resolution of 0.5 cm$^{-1}$, spectrally sampled at 0.25 cm$^{-1}$. IASI has a wavenumber-dependent radiometric noise ranging from 0.1 to 0.4 K for a reference blackbody at 280 K (Clerbaux et al., 2009), and more specifically around 0.15 K for HCOOH and 0.20 K for the CO spectral ranges (~1105 cm$^{-1}$ and ~2150 cm$^{-1}$, respectively).

The HCOOH and CO columns from IASI are used hereafter to determine the enhancement ratios of HCOOH. CO is chosen as reference due to its longer tropospheric lifetime (a few weeks to a few months depending on latitude and time of year) as compared to HCOOH. In our study we use CO as the reference and not $CO_2$ since variations in $CO_2$ concentration are difficult to measure with sufficient accuracy from IASI (Crevoisier et al., 2009).

### 2.2 The CO retrieval characteristics

The CO concentrations are retrieved from IASI using the FORLI-CO software (Hurtmans et al., 2012), which uses an optimal estimation method based on Rodgers (2000). The spectral range used for the retrieval is between 2143 and 2181.25 cm$^{-1}$. The CO total columns have been validated for different locations and atmospheric conditions (e.g. De Wachter et al., 2012; Kerzenmacher et al., 2012) and the comparison with other data have shown good overall agreement, even if some discrepancies were found within CO-enriched plumes (reaching 12% over the Arctic in summer, see Pommier et al., 2010; and reaching 17% in comparison with other IR sounders, see George et al., 2009). These data were also used previously to study biomass burning plumes (e.g. Turquety et al., 2009; Pommier et al., 2010; Krol et al., 2013; Whitburn et al., 2015).

In order to keep only the most reliable retrievals, the selected data used have a root-mean-square error lower than 2.7×10$^{-9}$ W/(cm$^2$ cm$^{-1}$ sr) and a bias ranging between -0.15 and 0.25×10$^{-9}$ as recommended in Hurtmans et al. (2012).

### 2.3 The HCOOH retrieval characteristics

The retrieval is based on the determination of the brightness temperature difference ($\Delta T_b$) between spectral channels with and without the signature of HCOOH. The reference channels used for the calculation of $\Delta T_b$ were chosen on both sides of the HCOOH Q-branch (1105 cm$^{-1}$), i.e., at 1103.0 and 1109.0 cm$^{-1}$. These $\Delta T_b$ were converted into total columns of HCOOH using conversion factors compiled in look-up tables. This simple and efficient retrieval method is described in more detail in Pommier et al. (2016).

As shown in Pommier et al. (2016), the vertical sensitivity of the IASI HCOOH total column ranges between 1 and 6 km. That study also showed that large HCOOH total columns were detected over biomass burning regions (e.g. Africa, Siberia) even if the largest values were found to be underestimated. This underestimation, which is less than 35% for the columns smaller than 2.5×10$^{16}$ molec/cm$^2$ (Pommier et al., 2016), will affect the enhancement ratios calculated in this work.

On the other hand, a large overestimation of the IASI HCOOH columns was shown in comparison with ground-based FTIR. This overestimation was larger for background columns (expected to reach 80% for a column close to 0.3×10$^{16}$ molec/cm$^2$), which can also impact our enhancement ratios.

## 3. MODIS

To identify the fire locations (hotspots), the fire product from MODIS on board the polar orbiting sun-synchronous NASA Terra and Aqua satellites (Justice et al., 2002; Giglio et al., 2006) are used. The Terra and Aqua satellites equatorial overpass times are ~10:30 (am and pm) and ~01:30 (am and pm) local time, respectively. Fire pixels are 1 km×1 km in size at nadir. For this work, we more specifically use the Global Monthly Fire Location Product (MCD14ML, Level 2, Collection 5)

developed by the University of Maryland (https://earthdata.nasa.gov/data/near-real-time-data/firms/active-fire-data#tab-content-6) which, for each detected fire pixel, includes the geographic location of the fire, the fire radiative power (FRP), the confidence in detection, and the acquisition date and time. The FRP provides a measure of fire intensity that is linked to the fire fuel consumption rate (e.g. Wooster et al., 2005). Only data presenting a high confidence percentage are used, i.e. higher than or equal to 80% as recommended in the MODIS user's guide (Giglio, 2013).

To characterize each MODIS hotspot by the type of fuel burned, the Global Mosaics of the standard MODIS land cover type data product (MCD12Q1) in the IGBP Land Cover Type Classification (Friedl et al., 2010; Channan et al., 2014) with a 0.5° × 0.5° horizontal resolution has also been used (http://glcf.umd.edu/data/lc/). As the annual variability in this product is limited (not shown) and since the period available (from 2001 to 2012) does not fully match the period of the IASI mission, only the data for 2012 have been used. Whitburn et al. (2017) have also used this MCD12Q1 product to determine their IASI-derived

$NH_3$ enhancement ratios by vegetation types.

## 4. Identifying fire-affected IASI observations

### 4.1 The selected areas

The determination of the biomass burning regions is based on the MODIS fire product. Figure 1 highlights the main areas that contributed to the biomass burning for the period between 2008 and 2014. Seven regions were selected for this work: Amazonia

(AMA, 5-15°S 40-60°W), corresponding mainly to the Brazilian Cerrado, Australia (AUS, 12-15°S 131-135°E), Northern Africa (NAF, 3-10°N 15-30°E), Southern Africa (SAF, 5-10°S 15-30°E), Southern East Asia (SEA, 18-27°N 96-105°E), India (IND, 15-27°N 75-88°E) and Siberia (SIB, 55-65°N 80-120°E). Among these regions, India and Siberia do not represent the most active regions in terms of number of fires. It seemed however important to also investigate them. One first reason for this is that Pommier et al. (2016) showed a misrepresentation of the fire emissions of HCOOH over India. Secondly, India also

encounters excess of acidity in rainwater, which could be partly attributed to biomass burning (e.g. Bisht et al., 2014). Concerning Siberia, this region and the surrounding areas experienced intense fires over some years, such as during the summer 2010 (Pommier et al., 2016; and R'honi et al. 2013 for the region close to Moscow). The classification of the vegetation from the MODIS product has also been used for a detailed analysis of the enhancement ratios for these regions (Fig. 1).

### 4.2 The IASI data used

For this work, both the daytime and nighttime IASI data were used. We have verified that using only the daytime retrievals did not change the results. Figure 2 presents the time-series of the monthly mean for the HCOOH and CO total columns over the seven selected regions. The number of fires and their FRP are also indicated. The variation in the total columns of HCOOH and CO matches relatively well with the variation of the number of fires. It is also worth noting that these variations in the total columns do not depend on the intensity of the fires as shown by Fig. 2 and by the scatterplots with the values characterizing

each fire as described below (not shown).

The monthly HCOOH and CO total columns are found to be highly correlated over the selected biomass burning regions (correlation coefficient, r, from 0.75 to 0.91), except over India (r=0.34) and Siberia (r=0.58). Over both regions, the impact of sources other than biomass burning is thus not negligible. Over India, the CO budget is influenced by long-range transport (e.g. Srinivas et al., 2016) and the anthropogenic emissions also have a large impact (e.g. Ohara et al., 2007). This could explain

why the variation in CO does not follow perfectly the variation in the number of fires and that the difference between the

background level and the CO peaks is less marked than for the HCOOH. Over Siberia, a temporal shift between the highest peaks for CO and for HCOOH is noticed for some years, such as for 2009, 2010 and 2011. For these years, the variation in CO does not follow the variation in the number of fires. The large region selected over Siberia is known also to be impacted by CO-enriched plumes transported from other regions, such as polluted air masses from China (e.g. Paris et al., 2008) or from Europe (e.g. Pochanart et al., 2003). These external influences interfere with the CO plumes originating from forest fires measured over this region.

Despite the overall good match between the number of fires and the variation in HCOOH and CO, we are not certain that the HCOOH and the CO were emitted solely by fires, and the discrimination between a natural and an anthropogenic origin for each compound is challenging. This assessment is particularly obvious for IND and SIB. To isolate the HCOOH and CO signals measured by IASI, potentially emitted by a fire, we decided to only use the data in the vicinity of each MODIS hotspot. To do so, we co-located the IASI data at 50 km around each MODIS pixel and between 0 and 5h from the time registered by MODIS for each detected fire, so that each MODIS pixel is associated with a mean value of HCOOH and CO total columns from IASI.

With these co-location criteria, good correlation coefficients, calculated by linear least-square fitting, are found between the HCOOH and CO total columns as shown in Table 1 (upper row). The smaller correlation coefficients, i.e., less than 0.7, are found for India, Australia, Siberia and Northern Africa. It is also important to note that the HCOOH and CO columns are better correlated for India and Siberia compared to the monthly time-series shown in Fig. 2. The three other regions present a large correlation, around 0.8. The high correlation suggests that IASI sampled the same biomass burning air mass for these compounds.

### 4.3 Importance of the meteorological conditions

As shown in earlier studies, the wind speed can have a large influence on the detection of tropospheric plumes of trace gases from space (e.g $NO_2$: Beirle et al., 2011; CO: Pommier et al., 2013; $SO_2$: Fioletov et al., 2015). We have chosen to assign a surface wind speed value for each MODIS hotspot. These meteorological fields were taken from the ECMWF (European Centre for Medium-Range Weather Forecasts) reanalysis data (http://data-portal.ecmwf.int/data/d/interim_full_daily) (Dee et al., 2011). The horizontal resolution of these fields is 0.125° on longitude and latitude with a 6h time step. As shown in Fig 3, the three regions where the HCOOH:CO correlations are found to be high (r close to 0.8), correspond to the regions where the surface wind speed was lower, i.e. for AMA, SEA and SAF. IND has also a low mean and median surface wind speed but the distribution of this surface wind speed over IND is more spread out than for AMA, SEA and SAF. It is also noteworthy that the IND and SEA regions are both characterized by higher wind speed at higher altitudes, i.e. for the pressure levels 650 and 450 hPa (not shown). This shows that the wind speed at higher altitudes has a lower influence on our correlations than the surface wind. When filtering out the data associated with a large surface wind (higher than 1.44 m/s), new correlations between the HCOOH and the CO total columns from IASI are calculated (Table 1 – lower row). This value of 1.44 m/s for the surface wind speed corresponds to the 25th percentile of the distribution of the three regions characterized by the lowest surface wind speed (Fig. 3).

The correlation coefficients, shown on the scatterplots in Fig. 4 and summarized in Table 1 (lower row), increase for all regions except over NAF, where the coefficient is found to be slightly lower than the previous correlation (Table 1 – upper row). The correlation coefficient is significantly improved over IND and SIB (Table 1 – lower row). These results confirm a robust correlation between the HCOOH and the CO total columns measured by IASI in the vicinity of each MODIS fire location.

### 5. Analysis of the data over the fire regions

### 5.1. Determination of the enhancement ratios

### 5.1.1 General analysis

Based on scatterplots in Fig. 4, an enhancement ratio can be calculated for each region. These enhancement ratios defined as $ER_{(HCOOH/CO)}$, correspond to the value of the slope $\partial[HCOOH]/\partial[CO]$ found in Fig 4. This technique to determine the $ER_{(HCOOH/CO)}$ is more reliable than using only the columns themselves, i.e. by estimating an $ER_{(HCOOH/CO)}$ for each measurement pair (HCOOH, CO). Indeed, to perform scatterplots helps to identify a common origin for HCOOH and CO. The values of the $ER_{(HCOOH/CO)}$ over each region are summarized in Table 2.

It is known that trace gas concentrations within smoke plumes can vary rapidly with time and are very sensitive to chemistry, so a comparison with previous work is always challenging, especially if these studies were performed over another altitude range, at a different location or period of the year.

A good agreement is however generally found with previous studies, even if it is important to keep in mind that an underestimation of our $ER_{(HCOOH/CO)}$ is possible due to the underestimation in the highest values of HCOOH as over the forest fires (see Section 2.3). On the other hand, the overestimation in the background column can also impact the calculation of our $ER_{(HCOOH/CO)}$. The effects of both biases are, however, limited since most of HCOOH total columns used in our analysis over the selected regions are higher than $0.3 \times 10^{16}$ molec/cm$^2$ and lower than $2.5 \times 10^{16}$ molec/cm$^2$ as explained in Section 2.3.

Nevertheless, in order to investigate the possible impact of the overestimation in the lower columns and the underestimation in the higher columns on the calculated ratios, a test was performed, by using only HCOOH columns with a thermal contrast larger than 10K. Indeed, the increase in the thermal contrast (i.e. the temperature difference between the surface and the first layer in the retrieved profile) leads to reducing the detection limit as shown in Pommier al. (2016). This enhancement of the detection level helps to minimize the bias in the retrieved total columns as explained in Crevoisier et al. (2014). For the analysis performed here, similar slopes and correlation coefficients were generally calculated, suggesting a negligible effect of this parameter on the biases. The only exception is an increase in $ER_{(HCOOH/CO)}$ over Siberia ($6.5 \times 10^{-3} \pm 0.19 \times 10^{-3}$ mol/mol when using only IASI measurements with TC above 10K against $4.4 \times 10^{-3}$ mol/mol $\pm 0.09 \times 10^{-3}$ in Table 2). It is worth noting that only 48% of the selected scenes remain over Siberia when applying this filter on thermal contrast (60% for SEA, 77% for AMA, 80% for SAF, 83% for AUS and NAF, and 89% for IND). This implies that the statistics on the fire emissions in the higher latitudes of Siberia is dominated by measurements with a low thermal contrast and thus with HCOOH total columns with higher uncertainties. However, the limited changes in slopes and correlation coefficients give us confidence that the results presented in Table 2 are representative.

### 5.1.2 Analysis over each region

A few backward trajectories (along 5 days, not shown) have been calculated for our hotspots with the online version of the HYSPLIT atmospheric transport and dispersion modeling system (Rolph, 2017). These trajectories, initialized at different altitudes, confirm a main origin close to the surface of our IASI fire-affected columns. It is however impossible to properly compare the origin of the air masses with previous studies as our studied period (2008-2014) or our studied fires do not necessarily match with plumes described in other publications. It is also difficult to estimate the age of our studied air masses by gathering the plumes during a 7-yr period and without an accurate knowledge of the altitude of the plumes.

When compared with other studies, the best agreement for the values presented in Table 2 is found over Southern Africa where the $ER_{(HCOOH/CO)}$ ($5 \times 10^{-3} \pm 0.13 \times 10^{-3}$ mol/mol) is similar to the value calculated by Vigouroux et al. (2012) and Coheur et al. (2007). It also agrees with the broad range of values of emission ratios ($EmR_{(HCOOH/CO)}$) referenced by Sinha et al. (2003). This result corroborates the relevance of the methodology used in this work over this region for the identification of fire-affected IASI columns close to the source. Vigouroux et al. (2012) sampled biomass burning outflow of Southern Africa, Coheur et al. (2007) calculated their $ER_{(HCOOH/CO)}$ in plumes observed over Tanzania in the upper troposphere while Sinha et al. (2003) did it within plumes over Zambia at the origin of the fire.

A few assumptions are needed in order to interpret our $ER_{(HCOOH/CO)}$ but the analysis given hereafter is only indicative since these previous studies did not measure the same plume as those presented in this work. Our $ER_{(HCOOH/CO)}$ is also calculated without making any distinction on the seasonal variation or on the type of biomass burning plumes sampled (e.g. emitted by a
savanna fire or by a forest fire). The analysis by biome is presented in Section 5.2. Since these $ER_{(HCOOH/CO)}$ from previous studies and the $EmR_{(HCOOH/CO)}$ from Sinha et al. (2003) agree with our $ER_{(HCOOH/CO)}$, and since HCOOH has a short lifetime, this may suggest that the selected plumes measured by IASI from 2008 to 2014 and those sampled by Vigouroux et al. (2012) and Coheur et al. (2007), encountered a limited secondary production or a low sink (deposition or reaction with OH in the troposphere during their transport). To quantify the role of the chemistry or of the deposition within the plumes, a modeling
work should be performed. This is however beyond the scope of this paper.

Another important point is that the decay of HCOOH is faster than for CO. As our $ER_{(HCOOH/CO)}$ is similar to the $ER_{(HCOOH/CO)}$ from the other studies and to the $EmR_{(HCOOH/CO)}$ given in Sinha et al. (2003), this could suggest that all these plumes (from our study, from Vigouroux et al. (2012) and Coheur et al. (2007)) are rapidly advected in the troposphere. Our $ER_{(HCOOH/CO)}$ differs however from the value in Rinsland et al. (2006) ($11.3\times10^{-3} \pm 7.6\times10^{-3}$ mol/mol), since our ratio is 55% lower. One possible
explanation is the multi-origin of the plumes studied by Rinsland et al. (2006), since, based on their backward trajectories, their plumes could be influenced by biomass burning originating from Southern Africa and/or from Southern America. The travel during the few days across the Atlantic Ocean may explain the change in their $ER_{(HCOOH/CO)}$.

It is worth noting that the ACE-FTS instrument used in their study works in a limb solar occultation mode. This means that the atmospheric density sampled by the instrument is larger than the one measured by the nadir geometry with IASI. However,
the difference in geometry cannot explain why we find an agreement with the ACE-FTS measurements values reported by Coheur et al. (2007) and a disagreement with those from Rinsland et al. (2006). Part of the difference could be associated with the difference in the assumptions used in both retrievals (e.g. the a priori).

The $ER_{(HCOOH/CO)}$ from our work is also 15% lower than the $EmR_{(HCOOH/CO)}$ in Yokelson et al. (2003) ($5.9\times10^{-3} \pm 2.2\times10^{-3}$ mol/mol) who calculated their value within plumes over Zambia, Zimbabwe and South Africa. With this difference we can
also suggest the presence of a sink of HCOOH within the plumes detected by IASI, or that this slight underestimation is simply related to the faster decay of HCOOH than the one of CO. Conversely, the $ER_{(HCOOH/CO)}$ retrieved from IASI is twice that of Chaliyakunnel et al. (2016) ($2.6\times10^{-3} \pm 0.3\times10^{-3}$ mol/mol). Chaliyakunnel et al. (2016) developed an approach allowing the determination of pyrogenic $ER_{(HCOOH/CO)}$ by reducing the impact of the mix with the ambient air. To do so, they calculated the $ER_{(HCOOH/CO)}$ in the vicinity of fire count from MODIS (averaged in a cell having the resolution of the GEOS-Chem model, i.e.
$2°\times 2.5°$) and they differentiated this value with a background $ER_{(HCOOH/CO)}$ defined by the concentrations distant from these fires. They concluded that their most reliable value on the amount of HCOOH produced from fire emissions was obtained for African fires.

Over Northern Africa, the calculated $ER_{(HCOOH/CO)}$ ($4\times10^{-3} \pm 0.19\times10^{-3}$ mol/mol) is 42% higher than the $ER_{(HCOOH/CO)}$ calculated in Chaliyakunnel et al. (2016) ($2.8\times10^{-3} \pm 0.4\times10^{-3}$ mol/mol). It is worth noting that NAF is the region characterized
by a scatterplot with the lowest correlation coefficient (Fig. 4).

A larger difference is found over Australia where the $ER_{(HCOOH/CO)}$ is $11.1\times10^{-3} \pm 1.37\times10^{-3}$ mol/mol. This $ER_{(HCOOH/CO)}$ is roughly the mean of both values reported by Paton-Walsh et al. (2005) and Chaliyakunnel et al. (2016). The difference between our work and that of Paton-Walsh (2005) may be explained by the different origin of the probed plume. In our case, the studied area corresponds to the Northern part of the Northern Territory with savanna-type vegetation (as shown in Section 5.2) while
Paton-Walsh et al. (2005) sampled bush fire plumes coming from the Eastern Coast of Australia, representative of Australian temperate forest. In the work done by Chaliyakunnel et al. (2016), a quite uncertain value is reported ($4.5\times10^{-3} \pm 5.1\times10^{-3}$ mol/mol), with an error larger than their $ER_{(HCOOH/CO)}$.

Over Amazonia, our $ER_{(HCOOH/CO)}$ ($7.3\times10^{-3} \pm 0.08\times10^{-3}$ mol/mol) is similar to the value given in Chaliyakunnel et al. (2016), who report, however, a larger bias over Amazonia. Over this region, our $ER_{(HCOOH/CO)}$ is higher than the one obtained by

González Abad et al. (2009) with ACE-FTS in the upper troposphere ($5.1 \times 10^{-3} \pm 1.5 \times 10^{-3}$ mol/mol). This difference with the study done by González Abad et al. (2009) may be explained by the difference in the altitude of the detection of the forest fire plume between IASI (mid-troposphere) and ACE-FTS (upper-troposphere) and thus by a difference in the ongoing chemistry within their respective sampled plumes. The geometry of the sampling (nadir vs limb) or the difference in the retrieval may also have an impact in the retrieved HCOOH.

The Siberian $ER_{(HCOOH/CO)}$ ($4.4 \times 10^{-3}$ mol/mol $\pm 0.09 \times 10^{-3}$) is found to be in good agreement with the wide range of values obtained by Tereszchuk et al. (2013) and Viatte et al. (2015). This $ER_{(HCOOH/CO)}$ is however lower than the ratios calculated by R'Honi et al. (2013) who focused on the extreme fire event that occurred in 2010.

For India and Southern East Asia, a comparison is not possible since no previous studies were reported. The comparison is performed next, based on the emission factors.

**5.2. Analysis based on the type of vegetation**

We have complemented our comparison of the enhancement ratios by comparing our ratios to emissions ratios calculated from emission factors found in literature. The main argument to perform such comparison is the lack of measurements of enhancement ratios over IND and SEA. Furthermore, such comparison from emission factors facilitates an analysis based on hypothesis about the type of vegetation burned.

Even if our methodology attempts to characterize the HCOOH emitted by biomass burning close to the source, our columns are probably not representative of the emission at the origin of the fire. The altitude of the sampling (mid-troposphere), even if an influence from the surface is shown, and the age of the plumes (at least a few hours) have a large impact on our enhancement ratios.

To perform a proper comparison with emission ratios, our enhancement ratios should be converted to emission ratios. To do so, it would be essential to take into account the decay of the compounds during the transport of the plume. However, due to the methodology used, i.e. averaging the data collected during a few hours (between 0 and 5h from the time registered by MODIS for each detected fire), the calculation of the decay of each compound is not possible. We therefore have compared our enhancement ratios to emission ratios and the comparison presented hereafter is mostly illustrative.

For both IND and SEA regions, the emission ratios have been calculated from the emission factors provided in Akagi et al. (2011). For the other regions, in addition to the values from Akagi et al. (2011), emission ratios were similarly calculated from emission factors given in other studies (listed in Table 3).

Based on the emission ratios, the emission factors are usually derived by this following Equation:

$$EF_{HCOOH} = EF_{CO} \times MW_{HCOOH}/MW_{CO} \times EmR_{(HCOOH/CO)} \qquad (1)$$

$EF_{HCOOH}$ is the emission factor for HCOOH; $EmR_{(HCOOH/CO)}$ is the molar emission ratio of HCOOH with respect to CO; $MW_{HCOOH}$ is the molecular weight of HCOOH; $MW_{CO}$ is the molecular weight of CO and $EF_{CO}$ is the emission factor for CO for dry matter, set to the value taken from Akagi et al. (2011).

Thus, based on equation (1), $EmR_{(HCOOH/CO)}$ values were calculated and compared with our $ER_{(HCOOH/CO)}$ (Table 3). In this calculation, the vegetation type characterizing each region is important. Some regions are composed of a mix of vegetation types as shown in Fig. 1. This is for example the case for AMA and SAF (e.g. White, 1981). Thus following the classification from Akagi et al. (2011), AMA and SAF are composed of tropical forest and savanna, characterized by an $EF_{CO}$ of $93 \pm 27$ g/kg and $63 \pm 17$ g/kg, respectively (Akagi et al., 2011). AUS and NAF correspond to a savanna fuel type. SIB is a boreal forest area with an $EF_{CO}$ of $127 \pm 45$ g/kg. Based also on the maps shown by Fig. 9 in Schreier et al. (2014), Fig. 13 in van der Werf, et al. (2010), the soil for IND is supposed to be mainly composed of cropland (agriculture), which is associated to an $EF_{CO}$ of $102 \pm 33$ g/kg, and probably also by extratropical forest which is characterized by an $EF_{CO}$ equal to $122 \pm 44$ g/kg and savanna with an $EF_{CO}$ of $63 \pm 17$ g/kg. The fuel type for SEA is supposed to be a mix of extratropical forest and savanna, with an $EF_{CO}$ of $122 \pm 44$

g/kg, and 63±17 g/kg, respectively. Cropland fuel type was also used since large agricultural biomass burning is occurring in this region (e.g. Duc et al., 2016).

In addition to the $EmR_{(HCOOH/CO)}$ calculated from the $EF_{HCOOH}$ given in the literature, a classification for our $ER_{(HCOOH/CO)}$ has also been done, based on the data from the MCD12Q1 product. As each hotspot is associated with a land cover value defined by the MCD12Q1 product, enhancement ratios by biome have been calculated. The limitations of this dataset are its coarse

resolution ($0.5° \times 0.5°$) and the lack of seasonal variation. It gives however a supplementary information on the type of fuel burned identified by MODIS. The corresponding $ER_{(HCOOH/CO)}$ are provided in Table 3. Only the values calculated from a scatterplot with a correlation coefficient higher than 0.4 are reported.

Despite the assumptions made, a fair agreement is found over Southern Africa. Our $ER_{(HCOOH/CO)}$ ($5\times10^{-3} \pm 0.13\times10^{-3}$ mol/mol) is indeed similar to the $EmR_{(HCOOH/CO)}$ calculated from Sinha et al. (2004) by using savanna fuel type, and the $ER_{(HCOOH/CO)}$ is

between both values calculated from Yokelson et al. (2003). This agreement is consistent since both previous studies sampled plumes emitted by savanna fires. Yokelson et al. (2003) and Sinha et al. (2004) both used the same sampling strategy. They sampled fire plumes by penetrating several minutes old plumes at relatively low altitude (up to 1.3 km for Sinha et al. (2004) and just above the flame front for Yokelson et al. (2003)). This agreement shows, as already described in the previous section, that our $ER_{(HCOOH/CO)}$ over Southern Africa is similar to their $EmR_{(HCOOH/CO)}$. It is also noteworthy, based on the MODIS land

cover type product, that all the studied hotspots are defined as savanna fires. On other hand, our $ER_{(HCOOH/CO)}$ is also similar to the $EmR_{(HCOOH/CO)}$ from Akagi et al. (2011) but for the tropical forest. A large underestimation compared to Rinsland et al. (2006) is found. This underestimation confirms the disagreement with their study already shown in Table 2.

Over Northern Africa, our $ER_{(HCOOH/CO)}$ is twice as large as the $EmR_{(HCOOH/CO)}$ provided by Akagi et al. (2011), probably due to the lower correlation found in our scatterplot. It is highly probable that our presumed fire-affected IASI columns are indeed

impacted by other air masses. The land classification based on the MODIS product also shows a diverse origin of the hotspots. For Amazonia, the calculated $ER_{(HCOOH/CO)}$ ($7.3\times10^{-3} \pm 0.08\times10^{-3}$ mol/mol) is close to the $EmR_{(HCOOH/CO)}$ given in Akagi et al. (2011) for the tropical forest ($5.2\times10^{-3}$ mol/mol), but it is three times higher than the values derived from Yokelson et al. (2007; 2008) for the same vegetation type. For the latter, it is worth noting that their factors have been corrected a posteriori (scaled down by a factor of 2.1), as described in their comment following the paper done by R'Honi et al. (2013) (see Yokelson et al.,

2013). As Yokelson et al. (2007; 2008) sampled the forest fire plumes by penetrating recent columns of smoke 200−1000m above the flame front, our $ER_{(HCOOH/CO)}$ may reflect a secondary production of HCOOH. This assuming secondary production is less substantial in comparison with the value from Akagi et al. (2011). The classification based on the type of fuel burned shows diverse origin of the fire plumes over Amazonia. Six biomes have been identified following the classification from the MCD12Q1 product.

Over Australia and over Siberia, the calculated $ER_{(HCOOH/CO)}$ is overestimated compared to the $EmR_{(HCOOH/CO)}$ given in Akagi et al. (2011) for a savanna fire and for a boreal forest, respectively. If our value for near-source estimation is correct, this would probably mean that the direct emission is underestimated (by 450% over Australia and by 60% over Siberia) or that a large secondary production of HCOOH from Australian and Siberian fires occurred. These hypotheses in biased emissions and/or secondary production need, however, to be verified with modeling studies. Over Australia, the difference is very large even if

the comparison done by Pommier al. (2016) with FTIR measurements showed that the lowest bias was found for the Australian site (-2% at Wollongong). Over Siberia, we also note that the region is characterized by fires emitted from six types of biome based on the classification from MODIS.

Finally, in this comparison, the studied plumes over India and Southern East Asia are certainly related to agricultural fires, even if the evergreen broadleaf forest seems to dominate in the MODIS land cover type product. This is strongly possible as

agricultural residue burning is prevalent in these regions (e.g. Kaskaoutis et al., 2014; Vadrevu et al., 2015). Over India and over Southern East Asia, our $ER_{(HCOOH/CO)}$ ($6.8\times10^{-3} \pm 0.44\times10^{-3}$ mol/mol for India and $5.8\times10^{-3} \pm 0.15\times10^{-3}$ mol/mol for Southern East Asia) are close to the value referenced by Akagi et al. (2011) for cropland fires ($6\times10^{-3}$ mol/mol). Since our

ER$_{(HCOOH/CO)}$ are close to EmR$_{(HCOOH/CO)}$ derived from the EF$_{HCOOH}$ in Akagi et al. (2011), this may suggests that the plumes studied over the 7-yr period correspond to fresh plumes where the chemistry or the physical sink is small. This is further supported by the fact that among the seven regions, IND and SEA have larger vertical velocity means close to the surface indicating a larger rising motion of the air masses (not shown).

In general, the ER$_{(HCOOH/CO)}$ calculated for a specific biome varies with the regions. This shows that the type of vegetation is not the only factor influencing the ER$_{(HCOOH/CO)}$. The ongoing chemistry within a plume is important and the age of the air masses impact the level of HCOOH and CO in the plumes.

## 6. Conclusions

Seven years of HCOOH data measured by IASI over seven different fire regions around the world were analyzed (AMA = Amazonia, AUS = Australia, IND = India, SEA = Southern East Asia, NAF = Northern Africa, SAF = Southern Africa, SIB = Siberia). By taking into account the surface wind speed and by characterizing each MODIS fire hotspot with a value of HCOOH and CO total columns, this work established enhancement ratios for the seven biomass burning areas and compared them to previously reported values found in literature.

The difficulties in performing such a comparison are associated with the difference in locations, altitude of the sampling and age of each fire plume studied in these previous publications. A fair agreement was however found for the enhancement ratios calculated in this work, in comparison with other studies, using satellite, airborne or FTIR measurements.

In agreement with previous studies, the plumes from Southern African savanna fires may reflect a limited secondary production or a limited sink occurring in the upper layers of the troposphere during their transport. Such assumptions, however, are difficult to verify by comparing individual plumes (from previous studies) with plumes gathered during a 7-yr period (from IASI), and remain speculative without a detailed modeling study. Plumes from agricultural fires over India and Southern East Asia probably correspond to fresh plumes as our ER$_{(HCOOH/CO)}$ based on the 7-yr IASI measurements are similar to the EmR$_{(HCOOH/CO)}$ calculated from emission factors provided by Akagi et al. (2011).

A very good agreement in ER$_{(HCOOH/CO)}$ was found over Amazonia, especially in comparison with the work done by Chaliyakunnel et al. (2016) who determined pyrogenic ER$_{(HCOOH/CO)}$.

Fires over Australia and over Siberia are probably underestimated in terms of direct emission or secondary production of HCOOH. The analysis over Australia is however complicated as our ER$_{(HCOOH/CO)}$ approximately corresponds to the mean of the values reported in Paton-Walsh et al. (2005) and in Chaliyakunnel et al. (2016); and it is also 450% higher than the EmR$_{(HCOOH/CO)}$ derived from Akagi et al. (2011). The underestimation by 60% over Siberia is consistent with conclusions given in R'Honi et al. (2013). The calculation of the ER$_{(HCOOH/CO)}$ by biome shows that Siberian plumes are related to the burning of six different vegetation classes. The underestimation reported is thus difficult to confirm without the use of a chemical transport model.

The values found over Northern Africa were the more difficult to interpret as this region is characterized by a poorer correlation between our fire-affected HCOOH and CO total columns.

Finally, the estimation of the ER$_{(HCOOH/CO)}$ calculated by the type of vegetation burned, as referenced in the MODIS product, varies with the regions. This shows that other parameters than the type of fuel burned also influence the ER$_{(HCOOH/CO)}$.

With these findings and by updating the enhancement ratios, an interesting modeling study could be performed to estimate a new tropospheric budget for HCOOH. This IASI data set may also be used in the future to study a single plume at different times to inform on the loss during transport. Further insight into the transport and chemistry may be gained by using IASI's capability to measure several fire species simultaneously, such as HCN or $C_2H_2$ (e.g. Duflot et al., 2015). This would be useful for the characterization of the chemistry ongoing in a fire plume outflow.

An inter-comparison with other space-borne instruments such as TES and ACE-FTS will be helpful to interpret the difference and the biases between the retrieved HCOOH columns and thus between their respective ER$_{(HCOOH/CO)}$.

**7. Data availability**

The IASI FORLI CO and HCOOH products are publicly available via the Aeris data infrastructure, using the following links: http://iasi.aeris-data.fr/CO/ and http://iasi.aeris-data.fr/HCOOH/.

**Acknowledgments**

The IASI mission is a joint mission of EUMETSAT and the Centre National d'Etudes Spatiales (CNES, France). The IASI L1 data are distributed in near real time by EUMETSAT through the EUMETCast system distribution. We thank the MODIS team for providing public access to fire products MCD14ML and the land cover type data product MCD12Q1. This MCD14ML MODIS data set was provided by the University of Maryland and NASA FIRMS operated by NASA/GSFC/ESDIS with funding provided by NASA/HQ. The authors thank S. Whitburn (ULB) for his help on the MODIS files. They also thank J. Hadji-Lazaro (LATMOS) and L. Clarisse (ULB) for preparing the IASI $\Delta T_b$ data set. The authors also acknowledge ECMWF for free access to the meteorological data.

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

**Table 1.** Upper row: Correlation coefficients between the HCOOH total columns and the CO total columns measured by IASI
for the period between 2008 and 2014 over the seven studied regions. Lower row: As upper row but with only MODIS fire hotspot having a surface wind speed lower than 1.44 m/s. Each IASI data is selected in an area of 50 km around the MODIS fire hotspot and up to 5h after the time recorded for each fire. The number of fires characterized by HCOOH and CO total columns is given in parenthesis.

|   | AMA | AUS | IND | SEA | SAF | NAF | SIB |
|---|---|---|---|---|---|---|---|
| r | 0.78 (13342) | 0.63 (1525) | 0.53 (1641) | 0.84 (1865) | 0.78 (12227) | 0.58 (21139) | 0.65 (22353) |
|   | 0.79 (4580) | 0.65 (93) | 0.65 (340) | 0.86 (528) | 0.80 (895) | 0.53 (1095) | 0.72 (2097) |





**Table 2.** Enhancement ratio of HCOOH relative to CO (mol/mol) with its standard deviation compared to enhancement ratios of HCOOH relative to CO and emissions ratios of HCOOH reported in the literature for the seven studied regions.

| Region | Enhancement Ratio to CO (mol/mol) – this work | Enhancement Ratio to CO (mol/mol) found in literature | Emission Ratio to CO (mol/mol) found in literature | Instrument used |
|---|---|---|---|---|
| AMA | $7.3\times10^{-3} \pm 0.08\times10^{-3}$ | $5.1\times10^{-3} \pm 1.5\times10^{-3}$ (González Abad et al., 2009)* | | ACE-FTS |
| | | $6.7\times10^{-3} \pm 2.1\times10^{-3}$ (Chaliyakunnel et al., 2016) | | TES |
| AUS | $11.1\times10^{-3} \pm 1.37\times10^{-3}$ | $4.5\times10^{-3} \pm 5.1\times10^{-3}$ (Chaliyakunnel et al., 2016) | | TES |
| | | $21.0\times10^{-3} \pm 10.0\times10^{-3}$ (Paton-Walsh et al., 2005)* | | Ground-based FTIR |
| IND | $6.8\times10^{-3} \pm 0.44\times10^{-3}$ | None | | - |
| SEA | $5.8\times10^{-3} \pm 0.15\times10^{-3}$ | None | | - |
| NAF | $4.0\times10^{-3} \pm 0.19\times10^{-3}$ | $2.8\times10^{-3} \pm 0.4\times10^{-3}$ (Chaliyakunnel et al., 2016) | | TES |
| SAF | $5.0\times10^{-3} \pm 0.13\times10^{-3}$ | $2.6\times10^{-3} \pm 0.3\times10^{-3}$ (Chaliyakunnel et al., 2016) | | TES |
| | | $4.6\times10^{-3} \pm 0.3\times10^{-3}$ (Vigouroux et al., 2012) | | Ground-based FTIR |
| | | $5.1\times10^{-3}$ (Coheur et al., 2007) | | ACE-FTS |
| | | $11.3\times10^{-3} \pm 7.6\times10^{-3}$ (Rinsland et al., 2006)* | | ACE-FTS |
| | | | $5.9\times10^{-3} \pm 2.2\times10^{-3}$ (Yokelson et al., 2003) | Airborne FTIR |
| | | | $5.1\text{-}8.7\times10^{-3}$ (Sinha al., 2003) | Airborne FTIR |
| SIB | $4.4\times10^{-3} \pm 0.09\times10^{-3}$ | $0.77\text{-}6.41\times10^{-3}$ (Tereszchuk et al., 2013) | | ACE-FTS |
| | | $2.69\text{-}15.93\times10^{-3}$ (Viatte et al., 2015) | | Ground-based FTIR |
| | | $10.0\text{-}32.0\times10^{-3}$ (R'honi et al., 2013) | | IASI |

* Their "emission ratios" are requalified as enhancement ratios in this study since their ratios were not measured at the origin
the fire emission but at high altitudes and/or further downwind of the fires.


**Table 3.** Enhancement ratio of HCOOH relative to CO (mol/mol) with its standard deviation and enhancement ratio of HCOOH relative to CO (mol/mol) by biome with its standard deviation calculated in this work. For each enhancement ratio by biome, the correlation coefficient and the number of MODIS hotspots are provided. The enhancement ratios are compared to emission ratios calculated from emission factors given in the literature for the seven studied regions. For the calculation of these emission ratios, the emission factors of CO for the corresponding fuel type given in Akagi et al. (2011) are used. Emission ratios of HCOOH relative to CO (mol/mol) calculated from the emission factors of HCOOH given in Akagi et al. (2011) for the corresponding fuel type are also provided.

| Region | Enhancement Ratio to CO (mol/mol) – this work | Enhancement Ratio to CO (mol/mol)[1] by biome[2] – this work | Emission Ratio to CO (mol/mol) calculated from $EF_{HCOOH}$ given in literature and using $EF_{CO}$ from Akagi et al. (2011) | Instrument used |
|---|---|---|---|---|
| AMA | $7.3 \times 10^{-3} \pm 0.08 \times 10^{-3}$ | $6.3 \times 10^{-3} \pm 0.22 \times 10^{-3}$ **(Evergreen Broadleaf forest, r=0.81, n = 454)**<br><br>$3.0 \times 10^{-3} \pm 0.81 \times 10^{-3}$ **(Open shrubland, r=0.91, n = 5)**<br><br>$7.0 \times 10^{-3} \pm 2.47 \times 10^{-3}$ **(Woody savanna, r=0.63, n = 14)**<br><br>$7.6 \times 10^{-3} \pm 0.09 \times 10^{-3}$ **(Savanna, r=0.79, n = 3909)**<br><br>$8.4 \times 10^{-3} \pm 0.39 \times 10^{-3}$ **(Grassland, r=0.88, n = 143)**<br><br>$4.6 \times 10^{-3} \pm 0.35 \times 10^{-3}$ **(Cropland, r=0.88, n = 54)** | $1.8 \times 10^{-3}$ – Tropical forest (Yokelson et al., 2007 ; 2008)[3]<br>$2.7 \times 10^{-3}$ – Savanna (Yokelson et al., 2007 ; 2008)[3]<br><br>$2.0 \times 10^{-3}$ – Savanna (Akagi et al., 2011)<br>$5.2 \times 10^{-3}$ – Tropical forest (Akagi et al., 2011) | Airborne FTIR (Yokelson et al., 2007) ; laboratory (Yokelson et al., 2008) catalogue |
| AUS | $11.1 \times 10^{-3} \pm 1.37 \times 10^{-3}$ | $5.7 \times 10^{-3} \pm 2.55 \times 10^{-3}$ **(Woody savanna, r=0.6, n = 11)**<br><br>$11.2 \times 10^{-3} \pm 1.49 \times 10^{-3}$ **(Savanna, r=0.65, n = 80)** | $2.0 \times 10^{-3}$ – Savanna (Akagi et al., 2011) | catalogue |
| IND | $6.8 \times 10^{-3} \pm 0.44 \times 10^{-3}$ | $6.6 \times 10^{-3} \pm 0.77 \times 10^{-3}$ **(Woody savanna, r=0.65, n = 103)**<br><br>$6.2 \times 10^{-3} \pm 0.62 \times 10^{-3}$ **(Cropland, r=0.58, n = 198)**<br><br>$8.8 \times 10^{-3} \pm 1.19 \times 10^{-3}$ **(Cropland/Natural vegetation mosaic, r=0.85, n =23)** | $2.0 \times 10^{-3}$ – Savanna (Akagi et al., 2011)<br>$2.7 \times 10^{-3}$ – Extratropical forest (Akagi et al., 2011)<br>$6.0 \times 10^{-3}$ – Cropland (Akagi et al., 2011) | catalogue |
| SEA | $5.8 \times 10^{-3} \pm 0.15 \times 10^{-3}$ | $5.6 \times 10^{-3} \pm 0.20 \times 10^{-3}$ **(Evergreen Broadleaf forest, r=0.83, n = 334)**<br><br>$6.3 \times 10^{-3} \pm 0.66 \times 10^{-3}$ **(Mixed forest, r=0.76, n = 70)**<br><br>$6.2 \times 10^{-3} \pm 0.38 \times 10^{-3}$ **(Woody savanna, r=0.86, n = 99)** | $2.0 \times 10^{-3}$ – Savanna (Akagi et al., 2011)<br>$2.7 \times 10^{-3}$ – Extratropical forest (Akagi et al., 2011)<br>$6.0 \times 10^{-3}$ – Cropland (Akagi et al., 2011) | catalogue |

| | | | | |
|---|---|---|---|---|
| | | $7.1\times10^{-3} \pm 0.99\times10^{-3}$ **(Cropland/Natural vegetation mosaic, r=0.84, n =23)** | | |
| NAF | $4.0\times10^{-3} \pm 0.19\times10^{-3}$ | $3.4\times10^{-3} \pm 0.63\times10^{-3}$ **(Evergreen Broadleaf forest, r=0.52, n = 78)** | $2.0\times10^{-3}$ – Savanna (Akagi et al., 2011) | catalogue |
| | | $3.3\times10^{-3} \pm 0.28\times10^{-3}$ **(Woody savanna, r=0.44, n = 569)** | | |
| | | $4.4\times10^{-3} \pm 0.29\times10^{-3}$ **(Savanna, r=0.59, n = 441)** | | |
| | | $22.6\times10^{-3} \pm 11.06\times10^{-3}$ **(Cropland/Natural vegetation mosaic, r=0.67, n = 7)** | | |
| SAF | $5.0\times10^{-3} \pm 0.13\times10^{-3}$ | **all hotspots are woody savanna** | $3.3\times10^{-3}$ – Tropical forest (Sinha et al., 2004)[4] $4.8\times10^{-3}$ – Savanna (Sinha et al., 2004)[4] | Airborne FTIR |
| | | | $4.1\times10^{-3}$ – Tropical forest (Yokelson et al., 2003) $6.0\times10^{-3}$ – Savanna (Yokelson et al., 2003) | Airborne FTIR |
| | | | $13\times10^{-3}$ – Tropical forest (Rinsland et al., 2006) $19.2\times10^{-3}$ – Savanna (Rinsland et al., 2006) | ACE-FTS |
| | | | $2.0\times10^{-3}$ – Savanna (Akagi et al., 2011) $5.2\times10^{-3}$ – Tropical forest (Akagi et al., 2011) | catalogue |
| SIB | $4.4\times10^{-3} \pm 0.09\times10^{-3}$ | $4.0\times10^{-3} \pm 0.31\times10^{-3}$ **(Evergreen Needleaf forest, r=0.63, n = 245)** | $2.7\times10^{-3}$ – Boreal forest (Akagi et al., 2011) | catalogue |
| | | $3.6\times10^{-3} \pm 0.16\times10^{-3}$ **(Deciduous Needleaf forest, r=0.66, n = 659)** | | |
| | | $3.4\times10^{-3} \pm 0.18\times10^{-3}$ **(Mixed forest, r=0.57, n = 759)** | | |
| | | $6.6\times10^{-3} \pm 0.48\times10^{-3}$ **(Open shrubland, r=0.76, n = 143)** | | |
| | | $6.0\times10^{-3} \pm 0.41\times10^{-3}$ **(Woody savanna, r=0.76, n = 155)** | | |
| | | $3.8\times10^{-3} \pm 0.65\times10^{-3}$ **(Permanent wetland, r=0.6, n = 63)** | | |

[1] Only the enhancement ratio to CO calculated from a scatterplot with a correlation coefficient higher than 0.4 are reported.
[2] The type of vegetation is defined by the land cover type data product (MCD12Q1).


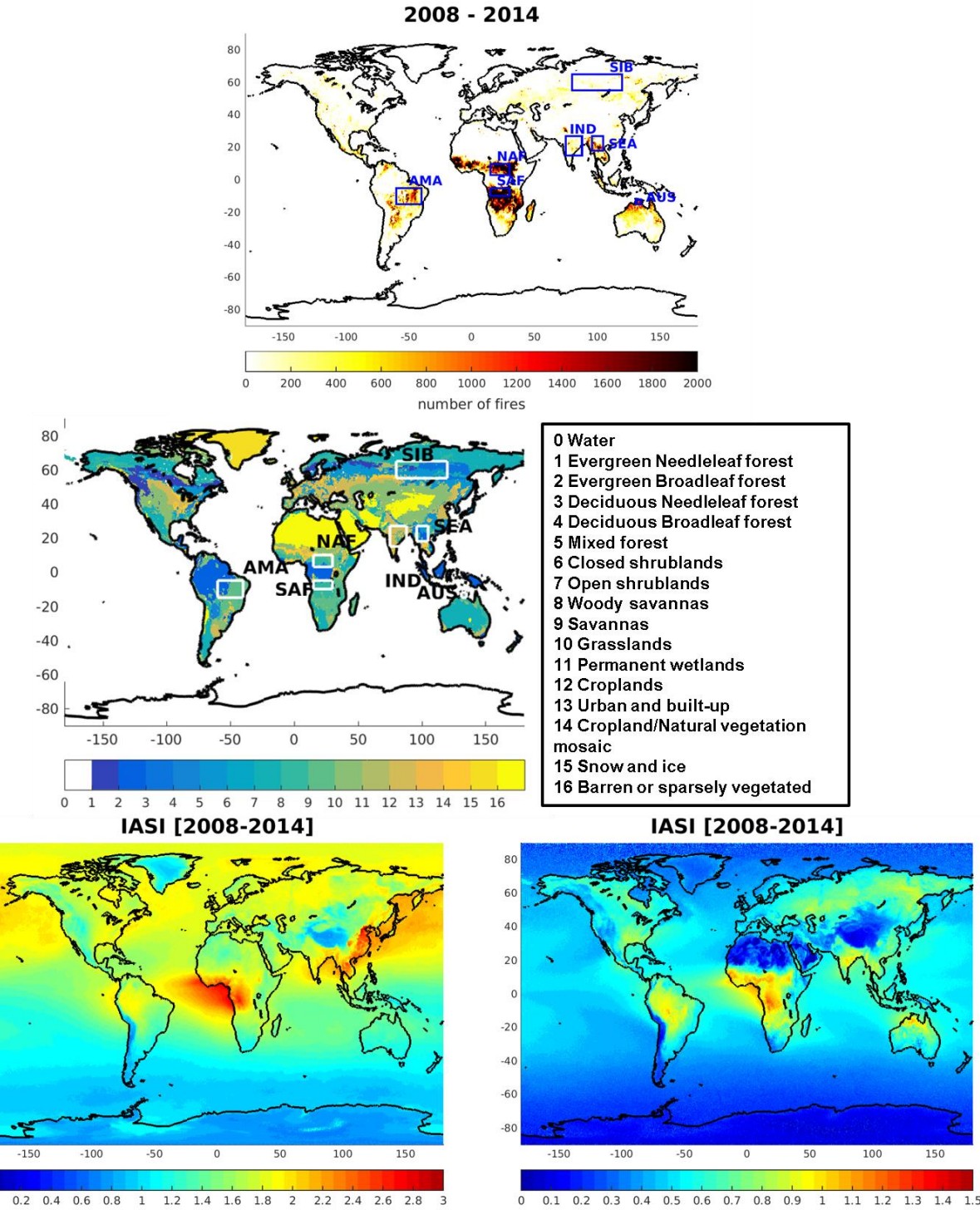

[3] The EF$_{HCOOH}$ were corrected based on the comment from Yokelson et al. (2013) (EF$_{HCOOH}$ used: 0.281 for Yokelson et al. (2007); 0.2767 for Yokelson et al. (2008)).

[4] The mean of both EF$_{HCOOH}$ values provided in Sinha et al. (2004) were used for our EmR$_{HCOOH/CO}$ calculation

**Figure 1: Top: Number of MODIS fire hotspots with a confidence percentage higher or equal to 80%, averaged on a 0.5°×0.5° grid, for the period between 2008 and 2014. The blue boxes are the regions studied in this work. Middle: Classification of the land cover type from MODIS on the same grid and highlighting the studied regions in white. Each number corresponds to the type of vegetation. Only the data between 64°S and 84°N are available. Bottom: The IASI CO total column distribution (left) and the IASI HCOOH total column distribution (right), averaged between 2008 and 2014 and on the same grid.**


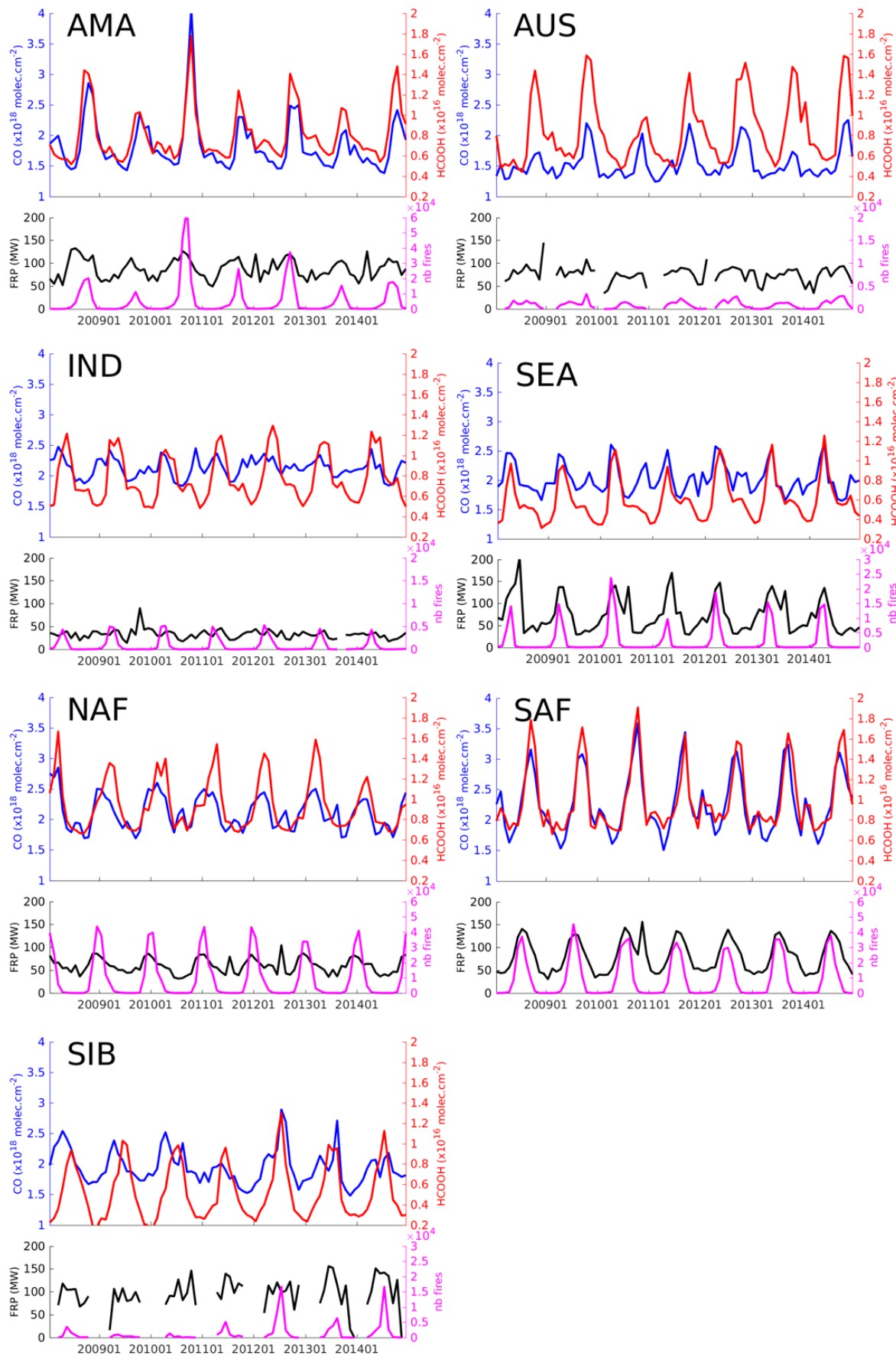

**Figure 2: Time-series from 2008 to 2014 of the monthly means of IASI CO (blue) and HCOOH (red) total columns in $10^{18}$ molec/cm$^2$ and in $10^{16}$ molec/cm$^2$, respectively, FRP (black) in MegaWatts and the number of fires (magenta) from MODIS over the seven regions (AMA=Amazonia, AUS=Australia, IND = India, SEA = Southern East Asia, NAF= Northern Africa, SAF= Southern Africa, SIB= Siberia).**


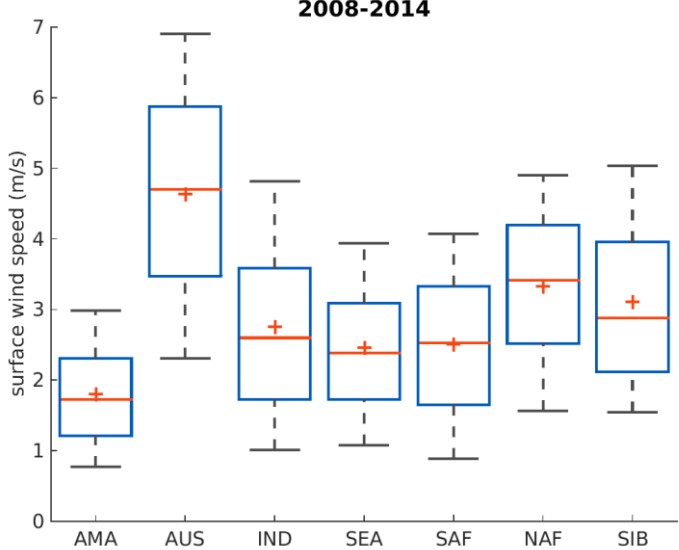

**Figure 3: Box and whisker plots showing mean (red central cross), median (red central line), and 25th and 75th percentile (blue box edges) of surface wind speed for each MODIS hotspot over the studied regions (AMA=Amazonia, AUS=Australia, IND = India, SEA = Southern East Asia, NAF= Northern Africa, SAF= Southern Africa, SIB= Siberia). The whiskers encompass values from 25th-1.5×(75th-25th) to the 75th+1.5× (75th-25th). This range of values corresponds to approximately 99.3% coverage if the data are normally distributed.**

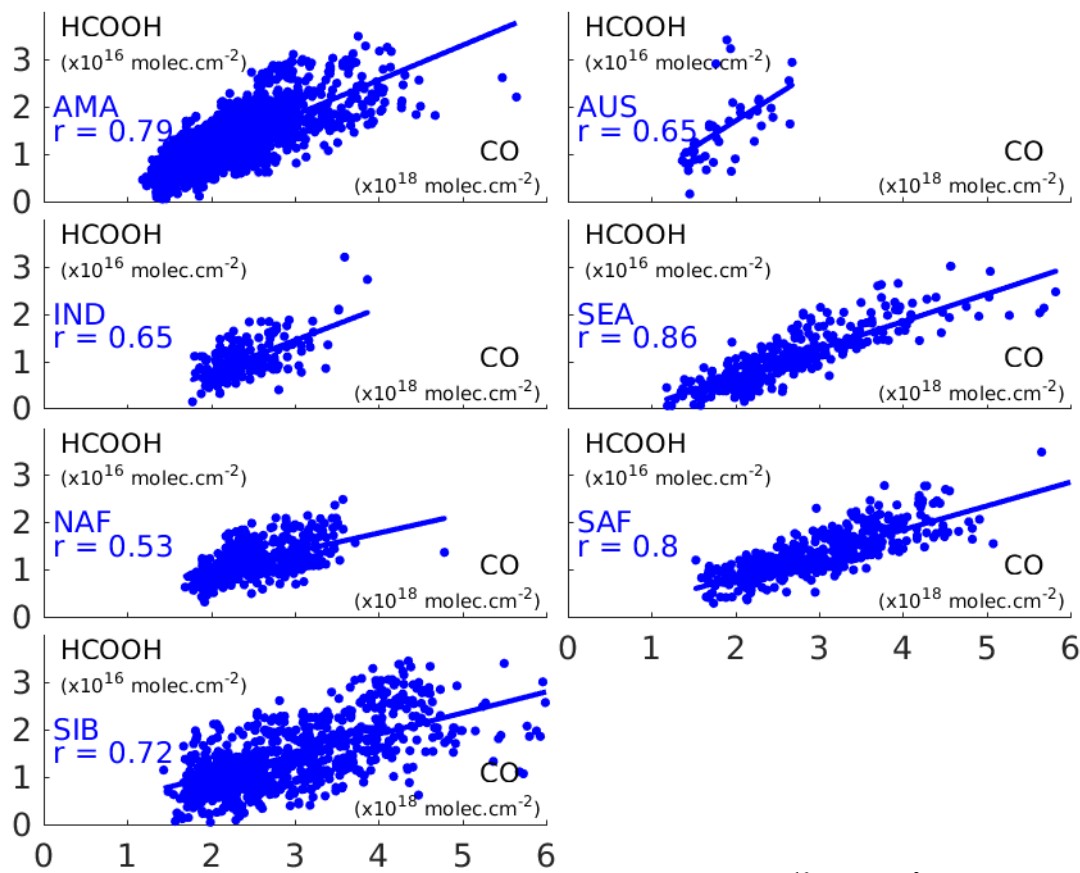

**Figure 4: Scatterplots between the IASI fire-affected HCOOH total columns (in $10^{16}$ molec/cm²) and the CO total columns (in $10^{18}$ molec/cm²) over the seven regions (AMA=Amazonia, AUS=Australia, IND = India, SEA = Southern East Asia, NAF= Northern Africa, SAF= Southern Africa, SIB= Siberia). The linear regression is represented by the blue line and the correlation coefficient is also provided for each region.**