# Peer review of "Possibility for an infrared sounder as IASI to document the HCOOH chemistry in biomass burning plumes"

_Atmospheric Chemistry and Physics, 2017_

## Referee Comment (RC1) · Anonymous Referee #1 · 5 Apr 2017

In this manuscript, Pommier et al., report a set of formic acid (HCOOH) enhancement ratios with respect to carbon monoxide (CO) derived from 2008 – 2014 IASI measurements. The authors pay special attention to 7 biomass burning regions comparing their estimates with previous studies. The comparisons show reasonable agreement. In the context of recent studies reporting large underestimations in the HCOOH atmospheric budget (i.e. Stavrakou et al. 2011) the IASI dataset can help to understand a fraction of the underestimation. However for publication in ACP I suggest the paper to undergo major revisions.

Abstract: With the evidence provided in the text the following sentence is not fully supported "The comparison with other studies highlights a possible underestimation by 60% of emission or a secondary production of HCOOH by Siberian forest fires while the studied fire plumes originating from Southern African savanna could suggests a limited secondary production of HCOOH or a limited sink." The differences in ER between different studies, need to be explained to support such conclusion.

Section 4.2: Figure 2 provides a qualitative analysis. HCOOH and CO concentrations apparently track MODIS fire counts. Working out correlations coefficients for CO, HCOOH and fire counts separately will help to address the origin of the air masses and what is the influence of the fire activity on them.

Sections 4.2 and 4.3: The authors try to isolate IASI retrievals influenced by biomass burning using MODIS and ECMWF data. While the definition of the biomass burning regions based in MODIS fire counts is clear, it is not clear to me how co-located IASI data are selected. Quoting the text: "To do so, we co-located the IASI data at 50 km around each MODIS pixel and between 0 and 5h for each detected fire, so that each MODIS pixel is associated with a value of HCOOH and CO total column from IASI". Further clarification is needed. All these questions are not answered in the description given in the text. For a MODIS pixel is it possible to have more than one IASI retrieval within 50 km? If so, the associated value for that MODIS fire is the average? MODIS has a resolution of 1km by 1km, a given retrieval can be accounted several times due to adjacent MODIS fire pixels. What does it mean 0 and 5 h for each detected fire? 5 hours ahead and 5 hours behind? With MODIS overpass times at 10:30am and 13:30am the night time IASI measurements 9:30pm will always be excluded. What is the influence of modifying the 50 km and 5 hour threshold in the results?

Surface ECMWF winds definitely increase the confidence of using only biomass burning affected IASI retrievals. However, the sensitivity of the IASI retrievals is highest between 1km and 6km. The authors should address the uncertainties introduced in the calculations due to transport vs. lofting of the air masses and influence of non-pyrogenic air masses in the IASI retrievals. This is particularly relevant for regions other than Equatorial Africa and South Africa were biomass burning signal is superimposed with other sources (Chaliyakunnel et al., 2016).

Table 1 and 2 can be combined in one single table.

Sections 5.1 & 5.2: What is the reason for the exception in Siberia where using only columns with a thermal contrast larger than 10K changed the ER from 6.5 mol/mol to 4.4 mol/mol.

Ground based FTIR, IASI, ACE-FTS, TES, and airborne FTIR are sensitive to different altitudes. The good agreement over Southern Africa can be linked with the distinctive burning season and air masses not containing other origins. That can explain why when ACE-FTS samples air masses that have travelled across the Atlantic Ocean (Risland et al., 2006) the ER are significant. Therefore, to extract quantitative

conclusions from the comparison exercise, it is necessary to have information about the origin of the air masses and the type of fuel burned. The authors can address these two issues using back trajectory model, for example Hysplit, and MODIS land surface type. As the manuscript stands now the discussion is mostly speculative.

Conclusions: As with the abstract "Fires over Australia and over Siberia are probably underestimated in terms of direct emission or secondary production of HCOOH. The analysis over Australia is however delicate as our $ER_{(HCOOH/CO)}$ approximately corresponds to the mean of the values reported in Paton-Walsh et al. (2005) and in Chaliyakunnel et al. (2016); and is also 450% higher than the $E_mR_{(HCOOH/CO)}$ derived from Akagi et al. (2011). The underestimation by 60% over Siberia is consistent with conclusions given in R'Honi et al., (2103)." a more detailed analysis is needed to link differences in ER with direct emission and secondary production.

Finally, IASI is also capable of measuring HCN a useful biomass burning tracer. It will be useful if the authors discussed the possibility of using it in future analysis.

Technical comments:

A revision of the English used could improve the transparency and clarity of the paper, particularly in the introduction.

Line 68, please include reference to Razavi et al., 2011 (first HCOOH retrievals from IASI).

Line 71, please include Gonzalez Abad et al., 2009 in ACE-FTS papers.

Line 98, please include citation about IASI $CO_2$ retrievals.

Line 118, correct typo (Pommier et al., 2016).

Line 141, actives to become active.

Line 206, should read "Both biases are however" instead of "Both biases is howeve"

Line 282, please specify which other studies.

Figure 2, include units in plots.

Figure 4, please include units in plots.

---

## Referee Comment (RC2) · Anonymous Referee #2 · 4 May 2017

General Comments

This manuscript presents IASI measurements of formic acid between 2008 and 2014, and uses these data to determine enhancement ratios from biomass burning emissions over seven regions. HCOOH and CO total columns, MODIS fire counts, and ECMWF surface wind speeds are combined to identify enhancements due to biomass burning. Correlations between HCOOH and CO total columns are used to calculate the enhancement ratio in each region. These results suggest that production of HCOOH by Siberian forest fires may be underestimated by 60%, and provide some insights into sources and sinks of HCOOH in other regions studied.

The manuscript provides a useful contribution to the field, but is somewhat qualitative

and speculative in places, as noted by the other reviewer. It also has many distracting grammatical errors and should be carefully reviewed and revised to correct these and to improve the clarity of the writing. I recommend publication in ACP after the comments below are addressed.

Specific Comments

Page 1, line 1 – The title is awkwardly phrased. Why just a "Possibility" for IASI to detect HCOOH in biomass burning plumes? "document" should be replaced by "measure" or "detect". A better title might be something like: "Detection of HCOOH from biomass burning plumes by the Infrared Atmospheric Sounding Interferometer"

Page 1, lines 25-27 – Make clear whether this underestimation for Siberian forest fires is in the IASI HCOOH or other studies or both. This seems rather speculative based on the results presented in the paper.

Page 1, lines 27-29 – Rewrite this last sentence for clarity.

Page 5, line 185 – Why is 1.44 m/s used as a threshold?

Page 6, lines 210-212 – Please clarify this discussion. It is not clear how a better detection limit "minimizes the bias with the lowest columns", nor what suggests "a negligible effect of the low column biases".

Page 6, para 3 – This is a long paragraph, written in a way that is hard to follow. Please revise for clarity. e.g., lines 224-228 – Explanations are also not clear here. Please explain why the results suggest that the plume "encountered a limited secondary production or a low sink as deposition or reaction with OH" and why the faster decay of HCOOH relative to CO, suggests rapid advection of the plumes. And line 237 – How would the impact of the difference in the geometry of sampling be accounted for in a proper comparison between ACE-FTS and IASI? Line 239 – Where were the plumes sampled by Yokelson et al.?

Page 7, lines 243-244 – What was the approach developed by Chaliyakunnel et al.

(2016) to determine pyrogenic ER(HCOOH/CO)? It is not clear what is meant "by reducing the impact of the mix with the ambient air".

Page 7, lines 269-271 – Revise this poorly written paragraph. It is not clear what is meant by either sentence.

Page 7, lines 275-279 – Why can't the decay be taken into account by considering the exponential decrease between emission and detection using relative lifetimes, e.g., Viatte et al. (2015) and references therein?

Sections 5.1 and 5.2 – Both sections discuss enhancement ratios and emission ratios, including comparisons with other studies, e.g., on page 8, there is additional discussion of ER although the title suggests that Section 5.2 is about EmR. These sections could be more clearly differentiated.

Page 9, lines 358-359 – Arguably, such an intercomparison could have been included in this study.

Technical Corrections

Page 1, line 19 – add comma after "(MODIS)"

Page 1, line 26 – add comma after "forest fires"

Page 1, line 34 – delete "for"

Page 2, line 46 – Rewrite this sentence. Not clear what is meant by "as on the oxidizing power…"

Page 2, line 55 – "hence depend on"

Page 2, line 67 – change "as with" to "including" or "such as"

Page 2, line 69 – delete "with the"

Page 2, line 70 – "Atmospheric Chemistry Experiment – Fourier Transform Spectrometer (ACE-FTS)"

Page 2, line 72 – I think this means "(MIPAS) limb instrument, which is sensitive to altitudes down to ∼10 km" (rather than only sensitive at 10 km)

Page 2, line 74 – "compared to ground-based and airborne"

Page 2, line 75 – "allows observation of remote regions"

Page 2, line 77 – "ratios of HCOOH relative to CO over"

Page 3, lines 93-94 – add space before K, as done for other units like km, cm-1, etc.

Page 3, line 97 – Isn't the lifetime of CO closer to two months than several weeks?

Page 3, line 113 – "in more detail"

Page 3, lines 117-118 – "which is less than 35% for total columns smaller than . . ."

Page 4, line 123 – "hotspots"

Page 4, line 123 – MODIS has already been defined

Page 4, line 129 – "which, for each detected fire pixel, includes the . . ."

Page 4, line 132 – Last sentence doesn't need to be a separate paragraph.

Page 4, line 141 – "most active in terms of actual fires but are still of interest. The first . . ." These four sentences about importance of biomass burning in India and Siberia could also be rewritten for clarity.

Page 4, line 144 – "over some years, such as during summer 2010"

Page 4, line 154 – "(correlation coefficient, r, from"

Page 4, line 155 – "the impact of sources other than biomass burning"

Page 4, line 156 – "also have"

Page 4, line 160 – "The large region selected over Siberia"

Page 4, line 161 – "other regions, such as polluted"

Page 5, line 170 – add comma after "criteria"

Page 5, line 171 – "in Table 1. The smaller correlation coefficients, i.e., less than 0.7, are found"

Page 5, line 172 – "the HCOOH and CO columns"

Page 5, line 178 – "assign" rather than "attribute" ?

Page 5, line 179 – ECMWF has already been defined

Page 5, line 182 – "(r close to 0.8)"

Page 5, line 183 – Clarify that the low mean and median refer to surface wind speed. Also rewrite the sentence on line 184 for clarity.

Page 5, line 186 and elsewhere through the manscript– "in Table 2" ? Does ACP accept Tab. as an abbreviation for Table?

Page 5, line 197 – "than using only the columns"

Page 5, line 198 – "for each measurement pair"

Page 6, line 201 – "so comparison with previous work is . . . over another"

Page 6, line 203 – should globally be generally?

Page 6, line 206 – "The effects of both biases are, however, limited"

Page 6, line 211 – "an improved [or a lower?] detection limit"

Page 6, line 222 – "same plume as"

Page 6, line 231 – trajectories

Page 6, line 235 – "reasons for the agreement"

Page 6, lines 241-242 – "Conversely, the . . . from IASI is twice that of Chaliyakunnel"

Page 7, line 247 – No need for a new paragraph here.

Page 7, line 248 – "worth noting"

Page 7, line 251 – "and that of Paton-Walsh (2005) may be explained"

Page 7, line 254 – quantify "quite uncertain"

Page 7, line 280 – "For both the IND"

Page 8, line 287 – Equation

Page 8, line 289 – "composed of tropical"

Page 8, line 292 – "composed of cropland"

Page 8, line 293 – "characterized by an"

Page 8, line 300 – "(2004) both used the same"

Page 8, line 307 – "twice the value" [also specify whether ER or EmR from Akagi]

Page 8, line 308 – "It is highly"

Page 8, line 314 – "forest fire plumes"

Page 9, line 336 – "difficulties . . . are"

Page 9, line 338 – "using satellite, airborne, or FTIR measurements"

Page 9, line 346 – "A very good agreement was found" in what? Specify.

Page 9, line 349 – Replace "delicate" with a better description.

Page 9, line 355 – "a modelling study could be"

Page 9, line 357 – times

Page 9, line 358 – "instruments such as"

Page 10, line 367 – Isn't IASI an instrument, not a mission?

Page 10, line 372 – "for free access"

Page 10, lines 385 and 387 – Inconsistent formatting of references for the same journal.

Page 17, caption line 2 – "over the 7 [seven] regions studied. IASI data are"

Page 17 – Tab. or Table ?

Pages 18 and 19, table headings – "HCOOH/CO Enhancement/Emission Ratio . . ." would be a better title

Page 18, Table 3 – Left justify all the table entries

Page 19, caption line 2 – "in the literature". Also, rewrite the full caption for conciseness and clarity, e.g., HCOOH/CO enhancement ratio, etc.

Page 20, caption line 3 – "column distribution . . . column distribution"

Page 21, Figure 2 and page 22, Figure 4– Preferable to have units on the y-axis labels, rather than just in the caption.

Page 22, Figure 3, caption line 4 – Clarify text describing the percentiles.

---

## Author Comment (AC1) · 9 Aug 2017

**Reviewer 1**

In this manuscript, Pommier et al., report a set of formic acid (HCOOH) enhancement ratios with respect to carbon monoxide (CO) derived from 2008 – 2014 IASI measurements. The authors pay special attention to 7 biomass burning regions comparing their estimates with previous studies. The comparisons show reasonable agreement. In the context of recent studies reporting large underestimations in the HCOOH atmospheric budget (i.e. Stavrakou et al. 2011) the IASI dataset can help to understand a fraction of the underestimation. However for publication in ACP I suggest the paper to undergo major revisions.

The authors would like to thank reviewer 1 for his comments which help to improve our study. We have tried to clarify the points raised by the reviewer and to answer all remarks. Our responses are written in blue in this document.
Sect. 5.1 and 5.2 have been largely rewritten and are not copied in the present replies in full, thus also please read the revised manuscript.

Abstract: With the evidence provided in the text the following sentence is not fully supported "The comparison with other studies highlights a possible underestimation by 60% of emission or a secondary production of HCOOH by Siberian forest fires while the studied fire plumes originating from Southern African savanna could suggests a limited secondary production of HCOOH or a limited sink." The differences in ER between different studies, need to be explained to support such conclusion.
This is a good remark. This statement has been deleted from the abstract, we have however added theses sentences (in bold) in the conclusion:
"The underestimation by 60% over Siberia is consistent with conclusions given in R'Honi et al. (2013). **The calculation of the $ER_{(HCOOH/CO)}$ by biome shows that Siberian plumes are related to the burning of six different vegetation classes. The underestimation reported is thus difficult to confirm without the use of a chemical transport model**."

We have also written in Section 5.2:
"These hypotheses in biased emissions and/or secondary production need, however, to be verified with modeling studies."

Section 4.2: Figure 2 provides a qualitative analysis. HCOOH and CO concentrations apparently track MODIS fire counts. Working out correlations coefficients for CO, HCOOH and fire counts separately will help to address the origin of the air masses and what is the influence of the fire activity on them.
The monthly means does not present a clear correlation as illustrated in Fig. 2 and explained by the sentence in the ACPD manuscript in Section 4.2 (lines 150-152) "It is also worth noting that these variations in the total columns do not depend on the intensity of the fires as shown by Fig. 2 and by the scatterplots with the values characterizing each fire as described below (not shown)."

The impact of the fire activity (FRP) was, however, studied. We did not find correlations between the intensity of each fire and the amount of CO or HCOOH (see Table below for each region) despite that the enhancements in the IASI-derived columns can confidently be attributed to fires. We have decided not to show these results in the manuscript.

| region | Criteria: time=[0 5h], r=50km | Criteria: time=[0 5h], r=50km, ws<1.44 m/s |
|---|---|---|
| NAF | r (FRP-HCOOH)=0.02
 r (FRP-CO)=0.1 | r (FRP-HCOOH)=-0.09
 r (FRP-CO)=0.13 |
| AMA | r (FRP-HCOOH)=0.03
 r (FRP-CO)=0.03 | r (FRP-HCOOH)=0.01
 r (FRP-CO)=-0.04 |
| AUS | r (FRP-HCOOH)=0.1
 r (FRP-CO)=0.08 | r (FRP-HCOOH)=0.01
 r (FRP-CO)=-0.08 |
| SIB | r (FRP-HCOOH)=0.11
 r (FRP-CO)=0.08 | r (FRP-HCOOH)=0.2
 r (FRP-CO)=0.21 |
| SAF | r (FRP-HCOOH)=0.04
 r (FRP-CO)=0.09 | r (FRP-HCOOH)=0.05
 r (FRP-CO)=0 |
| SEA | r (FRP-CO)=0.13
 r (FRP-CO)=0.11 | r (FRP-HCOOH)=0.24
 r (FRP-CO)=0.18 |
| IND | r (FRP-HCOOH)=0.04
 r (FRP-CO)=0.02 | r (FRP-HCOOH)=0.08
 r (FRP-CO)=0.1 |

Sections 4.2 and 4.3: The authors try to isolate IASI retrievals influenced by biomass burning using MODIS and ECMWF data. While the definition of the biomass burning regions based in MODIS fire counts is clear, it is not clear to me how co-located IASI data are selected. Quoting the text: "To do so, we co-located the IASI data at 50 km around each MODIS pixel and between 0 and 5h for each detected fire, so that each MODIS pixel is associated with a value of HCOOH and CO total column from IASI".

Further clarification is needed. All these questions are not answered in the description given in the text. For a MODIS pixel is it possible to have more than one IASI retrieval within 50 km? If so, the associated value for that MODIS fire is the average? MODIS has a resolution of 1km by 1km, a given retrieval can be accounted several times due to adjacent MODIS fire pixels. What does it mean 0 and 5 h for each detected fire? 5 hours ahead and 5 hours behind? With MODIS overpass times at 10:30am and 13:30am the night time IASI measurements 9:30pm will always be excluded. What is the influence of modifying the 50 km and 5 hour threshold in the results?

The criteria used correspond to a radius of 50 km around each MODIS hotspot and the time = [0 5h]. Then all the IASI data collocated around each MODIS hotspot were averaged.
To clarify this point we have changed the sentence in the manuscript to:
 "To do so, we co-located the IASI data at 50 km around each MODIS pixel and between 0 and 5h from the time registered by MODIS for each detected fire, so that each MODIS pixel is associated with a mean value of HCOOH and CO total columns from IASI".

The idea was to get a sufficient number of hotspots with a high correlation coefficient in order to be confident in the value of the slope $\partial[HCOOH]/\partial[CO]$.
We have chosen to use as temporal criterion +5 hours instead of ±5 hours in order to avoid selecting IASI data before the starting time of a fire. Different criteria on the time difference and the spatial mismatch have been tested in addition to those used in the paper. They are summarized in the Table below but are not shown in the manuscript:

Tab. Correlation coefficients between the HCOOH total columns and the CO total columns measured by IASI for the period between 2008 and 2014 over the seven studied regions, before the use of the wind speed criterion. The results from the criteria, h=[0 5h] and r=50km used in the paper are written in red.

| | AMA | AUS | IND | SEA | SAF | NAF | SIB |
|---|---|---|---|---|---|---|---|
| Criteria used in the paper | 0.78 (13342) | 0.63 (1525) | 0.53 (1641) | 0.84 (1865) | 0.78 (12227) | 0.58 (21139) | 0.65 (22353) |
| h=[0 5h] r=10km | 0.72 (1510) | 0.49 (114) | 0.64 (184) | 0.78 (312) | 0.69 (1965) | 0.42 (2752) | 0.39 (2426) |
| h=[0 10h] r=10km | 0.63 (3624) | 0.48 (1376) | 0.53 (1941) | 0.7 (10897) | 0.69 (12211) | 0.49 (5708) | 0.45 (6342) |
| h=[0 10 h] r=50km | 0.73 (32463) | 0.61 (12414) | 0.47 (20090) | 0.74 (87378) | 0.72 (124784) | 0.6 (58273) | 0.66 (46081) |
| h=[-5h +5h] r=50 km | 0.79 (253188) | 0.74 (33303) | 0.55 (42924) | 0.82 (123243) | 0.81 (504733) | 0.53 (439994) | 0.61 (78570) |

Concerning the MODIS overpass, it was an error in the text. The correct sentence and overpasses are:
"The Terra and Aqua satellites equatorial overpass times are ~10:30 (am and pm) and ~01:30 (am and pm) local time, respectively."

Surface ECMWF winds definitely increase the confidence of using only biomass burning affected IASI retrievals. However, the sensitivity of the IASI retrievals is highest between 1km and 6km. The authors should address the uncertainties introduced in the calculations due to transport vs. lofting of the air masses and influence of non-pyrogenic air masses in the IASI retrievals. This is particularly relevant for regions other than Equatorial Africa and South Africa were biomass burning signal is superimposed with other sources (Chaliyakunnel et al., 2016).

The impact of the air masses on the IASI CO and HCOOH retrievals represents a specific study which should be done but it is beyond the scope of this paper, even if it is a relevant question. However, to answer this question, we have analyzed the vertical velocity at 1000hPa provided by ECMWF and the wind speed at three levels: 825, 650 and 450 hPa. These fields have the same resolution of the data used in the paper, i.e. $0.125° \times .0.125°$ and a 6h time step. We have checked their impact on our scatterplots as done with the surface wind speed in the manuscript.

The question about the lofting of the air masses can be studied with the vertical velocity. We have plotted the distribution over the seven regions as in figure 3 of the paper and presented hereafter:

[Figure]

Fig. 1. Box and whisker plots showing mean (red central cross), median (red central line), and 25th and 75th percentile (blue box edges) of vertical velocity at 1000 hPa for each MODIS hotspot over the studied regions (AMA=Amazonia, AUS=Australia, IND = India, SEA = Southern East Asia, NAF= Northern Africa, SAF= Southern Africa, SIB= Siberia).

Since pressure decreases with height, negative values of the vertical velocity indicate rising motion in the atmosphere, and positive values indicate sinking air.

As shown by this figure 1, no clear relationship between the vertical velocity and the correlation found over the regions studied in our work is found. India showing a low correlation coefficient as presented in Tab. 1, does not show a particular difference with other regions. For example, SAF having a lower mean velocity and SEA having a higher mean velocity than IND, have a higher correlation coefficient than IND.

We can conclude that the vertical injection ("lofting of the air masses") has a negligible impact on our scatterplots.

It however suggests a higher rising motion of the air masses over IND and SEA as already stated in Sect. 5.2. We have decided to add this sentence (in bold):

"…this may suggests that the plumes studied over the 7-yr period correspond to fresh plumes where the chemistry or the physical sink is small. **This is further supported by the fact that among the seven regions, IND and SEA have larger vertical velocity means close to the surface indicating a larger rising motion of the air masses (not shown)**."

In order to estimate the impact of the long-range transport on our correlation coefficients, a similar methodology has been used with the wind at different pressure levels.

We chose 450, 650 and 825 hPa, corresponding approximately to 5.7, 3.1 and 1.4 km and the results are presented in Fig. 2 hereafter. These levels are within the range of vertical sensitivity of the IASI HCOOH retrieval, i.e. between 1 and 6 km. The regions showing the lowest correlation coefficient (Tab.1 in the manuscript) do not match with a high or low wind speed. It is however shown that a high mean and median wind speed are noticed over IND and SEA.

These distributions do not allow the identification of a clear influence of the long-range transport in our scatterplots.

[Figure]

Fig. 2. Box and whisker plots showing mean (red central cross), median (red central line), and 25th and 75th percentile (blue box edges) of wind speed at 450, 650 and 825hPa for each MODIS hotspot over the studied regions (AMA=Amazonia, AUS=Australia, IND = India, SEA = Southern East Asia, NAF= Northern Africa, SAF= Southern Africa, SIB= Siberia).

We have added these sentences in Sect. 4.3:
"It is also noteworthy that the IND and SEA regions are both characterized by higher wind speed at higher altitudes, i.e. for the pressure levels 650 and 450 hPa (not shown). This shows that the wind speed at higher altitudes has a lower influence on our correlations than the surface wind."

Table 1 and 2 can be combined in one single table.
It is a good suggestion. Both tables are now merged as below:

**Table 1** Upper row: Correlation coefficients between the HCOOH total columns and the CO total columns measured by IASI for the period between 2008 and 2014 over the seven studied regions. Lower row: As upper row but with only MODIS fire hotspot having a surface wind speed lower than 1.44 m/s. Each IASI data is selected in an area of 50 km around the MODIS fire hotspot and up to 5h after the time recorded for each fire. The number of fires characterized by HCOOH and CO total columns is given in parenthesis.

| | AMA | AUS | IND | SEA | SAF | NAF | SIB |
|---|---|---|---|---|---|---|---|
| r | 0.78 (13342) | 0.63 (1525) | 0.53 (1641) | 0.84 (1865) | 0.78 (12227) | 0.58 (21139) | 0.65 (22353) |
| | 0.79 (4580) | 0.65 (93) | 0.65 (340) | 0.86 (528) | 0.80 (895) | 0.53 (1095) | 0.72 (2097) |

Sections 5.1 & 5.2: What is the reason for the exception in Siberia where using only columns with a thermal contrast larger than 10K changed the ER from 6.5 mol/mol to 4.4 mol/mol.
More explanations are now given and the new paragraph is (the modifications are highlighted in bold):
"Nevertheless, in order to investigate the possible impact of the overestimation in the lower columns **and the underestimation in the higher columns** on the calculated ratios, a test was performed, by using only HCOOH columns with a thermal contrast larger than 10K. Indeed, the increase in the thermal contrast (i.e. the temperature difference between the surface and the first layer in the retrieved profile) leads to **reducing** the detection limit as shown in Pommier al. (2016)**. This enhancement of the detection level helps to minimize the bias in the retrieved total columns as explained in Crevoisier et al. (2014). For the analysis performed here, similar slopes and correlation coefficients were generally calculated**, suggesting a negligible effect of this parameter on the biases. The only exception is an increase in $ER_{(HCOOH/CO)}$ over **Siberia ($6.5 \times 10^{-3} \pm 0.19 \times 10^{-3}$ mol/mol when using only IASI**

**measurements with TC above 10K against 4.4×10⁻³ mol/mol ± 0.09×10⁻³ in Table 2). It is worth noting that only 48% of the selected scenes remain over Siberia when applying this filter on thermal contrast (60% for SEA, 77% for AMA, 80% for SAF, 83% for AUS and NAF, and 89% for IND). This implies that the statistics on the fire emissions in the higher latitudes of Siberia is dominated by measurements with a low thermal contrast and thus with HCOOH total columns with higher uncertainties. However, the limited changes in slopes and correlation coefficients give** us confidence that the results presented in Table 2 are **representative**."

Ground based FTIR, IASI, ACE-FTS, TES, and airborne FTIR are sensitive to different altitudes. The good agreement over Southern Africa can be linked with the distinctive burning season and air masses not containing other origins. That can explain why when ACE-FTS samples air masses that have travelled across the Atlantic Ocean (Risland et al., 2006) the ER are significant. Therefore, to extract quantitative conclusions from the comparison exercise, it is necessary to have information about the origin of the air masses and the type of fuel burned. The authors can address these two issues using back trajectory model, for example Hysplit, and MODIS land surface type. As the manuscript stands now the discussion is mostly speculative. It is a good remark from the reviewer.

As there are 9628 MODIS hotspots studied in this paper, it is difficult to calculate backward trajectories for each hotspot, especially as different altitude ranges need to be tested since the vertical sensitivity of IASI (CO & HCOOH) is located in the free troposphere.
In order to investigate this, a few tests were done to show the distinct origins of the air masses at different locations, periods of the year and altitudes of the plume. Specifically 5 hotspots have been chosen randomly for each region and 3 different altitudes have been used: 500 m (thus close to the surface), 2000 m and 5000 m (representing the free troposphere). In total, this represents 105 trajectories.
These trajectories show that the air masses initialized at 500 and 2000m are mainly influencing by air masses close to the surface, confirming an origin near the source of our IASI fire-affected columns. It also shows the difficulty to estimate the origin of the air masses without an accurate knowledge of the altitude of the plumes.
These trajectories were plotted through the HYSPLIT online service:

[Figure]

Fig 3. 5-day backward trajectories from HYSPLIT online service calculated at 3 altitudes: 500 m (red), 2000 m (blue) and 5000 m (green), for 5 hotspots chosen randomly over the 7 regions studied in the paper. The parameters characterizing each MODIS hotpots are summarized in the following table. The meteorological fields are from GDAS at 1°×1° horizontal resolution.

[Figure]

Fig 3. Continue

**Tab.** Characteristic of each MODIS hotspot used for the trajectories plotted in the previous figure. The dates, the time recorded by the instrument and the coordinates for each hotspot are written.

| AMA |
|---|
| 20100906  / hour (UTC)=13  / lat=-6.476 - lon=-49.71 |
| 20120220  / hour (UTC)=2  / lat=-9.937 - lon=-59.911 |
| 20120817  / hour (UTC)=13  / lat=-5.839 - lon=-46.987 |
| 20130921  / hour (UTC)=13  / lat=-11.777 - lon=-50.871 |
| 20131011  / hour (UTC)=13  / lat=-11.09 - lon=-48.229 |
| **AUS** |
| 20130910  / hour (UTC)=0  / lat=-12.841 - lon=132.327 |
| 20130913  / hour (UTC)=1  / lat=-12.916 - lon=132.323 |
| 20130927  / hour (UTC)=1  / lat=-12.215 - lon=131.169 |
| 20131002  / hour (UTC)=1  / lat=-13.696 - lon=131.59 |
| 20131003  / hour (UTC)=0  / lat=-13.428 - lon=133.844 |
| **IND** |
| 20090407  / hour (UTC)=4  / lat=21.503 - lon=82.645 |
| 20090407  / hour (UTC)=4  / lat=20.067 - lon=84.175 |
| 20110317  / hour (UTC)=5  / lat=21.572 - lon=77.328 |
| 20120416  / hour (UTC)=5  / lat=22.166 - lon=77.749 |
| 20140420  / hour (UTC)=5  / lat=23.17 - lon=75.544 |
| **SEA** |
| 20110414  / hour (UTC)=3  / lat=22.681 - lon=96.801 |
| 20120402  / hour (UTC)=3  / lat=19.435 - lon=101.908 |
| 20130315  / hour (UTC)=3  / lat=21.594 - lon=100.047 |
| 20130329  / hour (UTC)=3  / lat=21.426 - lon=98.744 |
| 20140314  / hour (UTC)=3  / lat=19.8 - lon=100.375 |

| NAF |
|---|
| 20080108  / hour (UTC)=8  / lat=5.672 - lon=28.741 |
| 20100116  / hour (UTC)=8  / lat=8.419 - lon=19.196 |
| 20110222  / hour (UTC)=8  / lat=6.148 - lon=29.351 |
| 20121205  / hour (UTC)=8  / lat=9.168 - lon=19.859 |
| 20140203  / hour (UTC)=8  / lat=6.024 - lon=17.002 |
| **SAF** |
| 20090622  / hour (UTC)=8  / lat=-7.339 - lon=21.495 |
| 20100815  / hour (UTC)=9  / lat=-5.351 - lon=15.48 |
| 20111004  / hour (UTC)=20  / lat=-5.129 - lon=27.708 |
| 20120718  / hour (UTC)=20  / lat=-8.822 - lon=16.974 |
| 20140805  / hour (UTC)=8  / lat=-8.721 - lon=18.288 |
| **SIB** |
| 20110602  / hour (UTC)=4  / lat=59.259 - lon=98.286 |
| 20110609  / hour (UTC)=4  / lat=57.882 - lon=99.823 |
| 20120623  / hour (UTC)=6  / lat=62.39 - lon=85.655 |
| 20130807  / hour (UTC)=3  / lat=64.98 - lon=118.109 |
| 20130819  / hour (UTC)=4  / lat=55.39 - lon=106.945 |

To reply to this point, we have added the following sentences (in bold) in the text:
**"A few backward trajectories (along 5 days, not shown) have been calculated for our hotspots with the online version of the HYSPLIT atmospheric transport and dispersion modeling system (Rolph, 2017). These trajectories, initialized at different altitudes, confirm a main origin close to the surface of our IASI fire-affected columns. It is however impossible to properly compare the origin of the air masses with previous studies as our studied period (2008-2014) or our studied fires do not necessarily match with plumes described in other publications. It is also difficult to estimate the age of our studied air masses by gathering the plumes during a 7-yr period and without an accurate knowledge of the altitude of the plumes."**

And:

"One possible explanation is the multi-origin of the plumes studied by Rinsland et al. (2006), since, based on their backward trajectories, their plumes could be influenced by biomass burning originating from Southern Africa and/or from Southern America. **The travel during the few days across the Atlantic Ocean may explain the change in their $ER_{(HCOOH/CO)}$."**

With the corresponding reference:
Rolph, G.D.: Real-time Environmental Applications and Display sYstem (READY) Website (http://www.ready.noaa.gov). NOAA Air Resources Laboratory, College Park, MD, 2017.

About the type of fuel burned, thanks to the reviewer, we have discovered that such information was available from the MODIS products.
Now, in Section 3. MODIS we have added this paragraph:
"To characterize each MODIS hotspot by the type of fuel burned, the Global Mosaics of the standard MODIS land cover type data product (MCD12Q1) in the IGBP Land Cover Type Classification (Friedl et al., 2010; Channan et al., 2014) with a $0.5° \times 0.5°$ horizontal resolution has also been used (http://glcf.umd.edu/data/lc/). As the annual variability in this product is

limited (not shown) and since the period available (from 2001 to 2012) does not fully match the period of the IASI mission, only the data for 2012 have been used. Whitburn et al. (2017) have also used this MCD12Q1 product to determine their IASI-derived NH$_3$ enhancement ratios by vegetation types."

With the corresponding references:

Channan, S., Collins, K., and Emanuel, W. R., Global mosaics of the standard MODIS land cover type data. University of Maryland and the Pacific Northwest National Laboratory, College Park, Maryland, USA, 2014.

and

Friedl, M. A., Sulla-Menashe, D., Tan, B., Schneider, A., Ramankutty, N., Sibley, A. and Huang, X., MODIS Collection 5 global land cover: Algorithm refinements and characterization of new datasets, 2001-2012, Collection 5.1 IGBP Land Cover,  Remote Sensing of Environment, 114 , 168–182, doi:10.1016/j.rse.2009.08.016, 2010.

and

Whitburn, S., Van Damme, M., Clarisse, L., Hurtmans, D., Clerbaux, C., and Coheur, P.-F.: IASI-derived NH3 enhancement ratios relative to CO for the tropical biomass burning regions, Atmos. Chem. Phys. Discuss., https://doi.org/10.5194/acp-2017-331, in review, 2017.

We have also added this sentence in Section 4.2:

"The classification of the vegetation from the MODIS product has also been used for a detailed analysis of the enhancement ratios for these regions (Fig. 1)."

And Fig. 1 has been modified as below:

[Figure]

**Figure 1: Top: Number of MODIS fire hotspots with a confidence percentage higher or equal to 80%, averaged on a 0.5°×0.5° grid, for the period between 2008 and 2014. The blue boxes are the regions studied in this work. Middle: Classification of the land cover type from MODIS on the same grid and highlighting the studied regions in white. Each number corresponds to the type of vegetation. Only the data between 64°S and 84°N are available. Bottom: The IASI CO total column distribution (left) and the IASI HCOOH total column distribution (right), averaged between 2008 and 2014 and on the same grid.**

Section 5.2 was also rewritten and now named "5.2. Analysis based on the type of vegetation" since ER (HCOOH/CO) by type of vegetation were also added in Table 3 as below:

**Table 3.** Enhancement ratio of HCOOH relative to CO (mol/mol) with its standard deviation and enhancement ratio of HCOOH relative to CO (mol/mol) by biome with its standard deviation calculated in this work. For each enhancement ratio by biome, the correlation coefficient and the number of MODIS hotspots are provided. The enhancement ratios are compared to emission ratios calculated from emission factors given in the literature for the seven studied regions. For the calculation of these emission ratios, the emission factors of CO for the corresponding fuel type given in Akagi et al. (2011) are used. Emission ratios of HCOOH relative to CO (mol/mol) calculated from the emission factors of HCOOH given in Akagi et al. (2011) for the corresponding fuel type are also provided.

| Region | Enhancement Ratio to CO (mol/mol) – this work | Enhancement Ratio to CO (mol/mol)[1] by biome[2] – this work | Emission Ratio to CO (mol/mol) calculated from $EF_{HCOOH}$ given in literature and using $EF_{CO}$ from Akagi et al. (2011) | Instrument used |
|---|---|---|---|---|
| AMA | $7.3\times10^{-3} \pm 0.08\times10^{-3}$ | $6.3\times10^{-3} \pm 0.22\times10^{-3}$ (**Evergreen Broadleaf forest, r=0.81, n = 454**) | $1.8\times10^{-3}$ – Tropical forest (Yokelson et al., 2007 ; 2008)[3] | Airborne FTIR (Yokelson et al., 2007) ; laboratory (Yokelson et al., 2008) |
| | | $3.0\times10^{-3} \pm 0.81\times10^{-3}$ (**Open shrubland, r=0.91, n = 5**) | $2.7\times10^{-3}$ – Savanna (Yokelson et al., 2007 ; 2008)[3] | |
| | | $7.0\times10^{-3} \pm 2.47\times10^{-3}$ (**Woody savanna, r=0.63, n = 14**) | $2.0\times10^{-3}$ – Savanna (Akagi et al., 2011) $5.2\times10^{-3}$ – Tropical forest (Akagi et al., 2011) | catalogue |
| | | $7.6\times10^{-3} \pm 0.09\times10^{-3}$ (**Savanna, r=0.79, n = 3909**) | | |
| | | $8.4\times10^{-3} \pm 0.39\times10^{-3}$ (**Grassland, r=0.88, n = 143**) | | |
| | | $4.6\times10^{-3} \pm 0.35\times10^{-3}$ (**Cropland, r=0.88, n = 54**) | | |
| AUS | $11.1\times10^{-3} \pm 1.37\times10^{-3}$ | $5.7\times10^{-3} \pm 2.55\times10^{-3}$ (**Woody savanna, r=0.6, n = 11**) | $2.0\times10^{-3}$ – Savanna (Akagi et al., 2011) | catalogue |
| | | $11.2\times10^{-3} \pm 1.49\times10^{-3}$ (**Savanna, r=0.65, n = 80**) | | |
| IND | $6.8\times10^{-3} \pm 0.44\times10^{-3}$ | $6.6\times10^{-3} \pm 0.77\times10^{-3}$ (**Woody savanna, r=0.65, n = 103**) | $2.0\times10^{-3}$ – Savanna (Akagi et al., 2011) $2.7\times10^{-3}$ – Extratropical forest (Akagi et al., 2011) | catalogue |
| | | $6.2\times10^{-3} \pm 0.62\times10^{-3}$ (**Cropland, r=0.58, n = 198**) | $6.0\times10^{-3}$ – Cropland (Akagi et al., 2011) | |
| | | $8.8\times10^{-3} \pm 1.19\times10^{-3}$ (**Cropland/Natural vegetation mosaic, r=0.85, n =23**) | | |
| SEA | $5.8\times10^{-3} \pm 0.15\times10^{-3}$ | $5.6\times10^{-3} \pm 0.20\times10^{-3}$ (**Evergreen Broadleaf forest, r=0.83, n = 334**) | $2.0\times10^{-3}$ – Savanna (Akagi et al., 2011) $2.7\times10^{-3}$ – Extratropical forest (Akagi et al., 2011) $6.0\times10^{-3}$ – Cropland (Akagi et al., 2011) | catalogue |

| | | | | |
|---|---|---|---|---|
| | | **6.3×10$^{-3}$ ± 0.66×10$^{-3}$ (Mixed forest, r=0.76, n = 70)** | | |
| | | **6.2×10$^{-3}$ ± 0.38×10$^{-3}$ (Woody savanna, r=0.86, n = 99)** | | |
| | | **7.1×10$^{-3}$ ± 0.99×10$^{-3}$ (Cropland/Natural vegetation mosaic, r=0.84, n =23)** | | |
| NAF | **4.0×10$^{-3}$ ± 0.19×10$^{-3}$** | **3.4×10$^{-3}$ ± 0.63×10$^{-3}$ (Evergreen Broadleaf forest, r=0.52, n = 78)** | 2.0×10$^{-3}$ – Savanna (Akagi et al., 2011) | catalogue |
| | | **3.3×10$^{-3}$ ± 0.28×10$^{-3}$ (Woody savanna, r=0.44, n = 569)** | | |
| | | **4.4×10$^{-3}$ ± 0.29×10$^{-3}$ (Savanna, r=0.59, n = 441)** | | |
| | | **22.6×10$^{-3}$ ± 11.06×10$^{-3}$ (Cropland/Natural vegetation mosaic, r=0.67, n = 7)** | | |
| SAF | **5.0×10$^{-3}$ ± 0.13×10$^{-3}$** | **all hotspots are woody savanna** | 3.3×10$^{-3}$ – Tropical forest (Sinha et al., 2004)[4] 4.8×10$^{-3}$ – Savanna (Sinha et al., 2004)[4] | Airborne FTIR |
| | | | 4.1×10$^{-3}$ – Tropical forest (Yokelson et al., 2003) 6.0×10$^{-3}$ – Savanna (Yokelson et al., 2003) | Airborne FTIR |
| | | | 13×10$^{-3}$ – Tropical forest (Rinsland et al., 2006) 19.2×10$^{-3}$ – Savanna (Rinsland et al., 2006) | ACE-FTS |
| | | | 2.0×10$^{-3}$ – Savanna (Akagi et al., 2011) 5.2×10$^{-3}$ – Tropical forest (Akagi et al., 2011) | catalogue |
| SIB | **4.4×10$^{-3}$ ± 0.09×10$^{-3}$** | **4.0×10$^{-3}$ ± 0.31×10$^{-3}$ (Evergreen Needleaf forest, r=0.63, n = 245)** | 2.7×10$^{-3}$ – Boreal forest (Akagi et al., 2011) | catalogue |
| | | **3.6×10$^{-3}$ ± 0.16×10$^{-3}$ (Deciduous Needleaf forest, r=0.66, n = 659)** | | |
| | | **3.4×10$^{-3}$ ± 0.18×10$^{-3}$ (Mixed forest, r=0.57, n = 759)** | | |

| |
|---|
| **6.6×10$^{-3}$ ± 0.48×10$^{-3}$ (Open shrubland, r=0.76, n = 143)** |
| **6.0×10$^{-3}$ ± 0.41×10$^{-3}$ (Woody savanna, r=0.76, n = 155)** |
| **3.8×10$^{-3}$ ± 0.65×10$^{-3}$ (Permanent wetland, r=0.6, n = 63)** |

[1] Only the enhancement ratio to CO calculated from a scatterplot with a correlation coefficient higher than 0.4 are reported.

[2] The type of vegetation is defined by the land cover type data product (MCD12Q1).

[3] The EF$_{HCOOH}$ were corrected based on the comment from Yokelson et al. (2013) (EF$_{HCOOH}$ used: 0.281 for Yokelson et al. (2007); 0.2767 for Yokelson et al. (2008)).

[4] The mean of both EF$_{HCOOH}$ values provided in Sinha et al. (2004) were used for our EmR$_{HCOOH/CO}$ calculation

We have added these sentences at the end of the Section 5.2:

"In addition to the EmR$_{(HCOOH/CO)}$ calculated from the EF$_{HCOOH}$ given in the literature, a classification for our ER$_{(HCOOH/CO)}$ has also been done, based on the data from the MCD12Q1 product. As each hotspot is associated with a land cover value defined by the MCD12Q1 product, enhancement ratios by biome have been calculated. The limitations of this dataset are its coarse resolution ($0.5° \times 0.5°$) and the lack of seasonal variation. It gives however a supplementary information on the type of fuel burned identified by MODIS. The corresponding ER$_{(HCOOH/CO)}$ are provided in Table 3. Only the values calculated from a scatterplot with a correlation coefficient higher than 0.4 are reported."

And

"In general, the ER$_{(HCOOH/CO)}$ calculated for a specific biome varies with the regions. This shows that the type of vegetation is not the only factor influencing the ER$_{(HCOOH/CO)}$. The ongoing chemistry within a plume is important and the age of the air masses impact the level of HCOOH and CO in the plumes."

We have also added these sentences in the abstract:

"An additional classification of the enhancement ratios by type of fuel burned is also provided, showing a diverse origin of the plumes sampled by IASI, especially over Amazonia and Siberia. The variability in the enhancement ratios by biome over the different regions show that the levels of HCOOH and CO do not only depend on the fuel types."

And in the conclusion:

"Finally, the estimation of the ER$_{(HCOOH/CO)}$ calculated by the type of vegetation burned, as referenced in the MODIS product, varies with the regions. This shows that other parameters than the type of fuel burned also influence the ER$_{(HCOOH/CO)}$."

Conclusions: As with the abstract "Fires over Australia and over Siberia are probably underestimated in terms of direct emission or secondary production of HCOOH. The analysis over Australia is however delicate as our ER (HCOOH/CO) approximately corresponds to the mean of the values reported in Paton- Walsh et al. (2005) and in Chaliyakunnel et al. (2016); and is also 450% higher than the E m R (HCOOH/CO) derived from Akagi et al. (2011). The underestimation by 60% over Siberia is consistent with conclusions given in R'Honi et al., (2103)." a more detailed analysis is needed to link differences in ER with direct emission and secondary production.

It is correct. See our responses to your first comment (abstract).

Finally, IASI is also capable of measuring HCN a useful biomass burning tracer. It will be useful if the authors discussed the possibility of using it in future analysis.
It is a good remark. This sentence has been modified (in bold) in the conclusion:
"This IASI data set may also be used in the future to study a single plume at different times **to inform on the loss during transport. Further insight into the transport and chemistry may be gained by using IASI's capability to measure several fire species simultaneously, such as HCN or $C_2H_2$ (e.g. Duflot et al., 2015**)."

The corresponding reference has also been added:
Duflot, V., Wespes, C., Clarisse, L., Hurtmans, D., Ngadi, Y., Jones, N., Paton-Walsh, C., Hadji-Lazaro, J., Vigouroux, C., De Mazière, M., Metzger, J.-M., Mahieu, E., Servais, C., Hase, F., Schneider, M., Clerbaux, C., and Coheur, P.-F.: Acetylene ($C_2H_2$) and hydrogen cyanide (HCN) from IASI satellite observations: global distributions, validation, and comparison with model, Atmos. Chem. Phys., 15, 10509-10527, doi:10.5194/acp-15-10509-2015, 2015.

Technical comments:
A revision of the English used could improve the transparency and clarity of the paper, particularly in the introduction.
It has been done.

Line 68, please include reference to Razavi et al., 2011 (first HCOOH retrievals from IASI).
The reference has been added.

Line 71, please include Gonzalez Abad et al., 2009 in ACE-FTS papers.
The reference has been added.

Line 98, please include citation about IASI CO2 retrievals.
This following reference has been added:
Crevoisier, C., Chédin, A., Matsueda, H., Machida, T., Armante, R., and Scott, N. A.: First year of upper tropospheric integrated content of $CO_2$ from IASI hyperspectral infrared observations, Atmos. Chem. Phys., 9, 4797-4810, doi:10.5194/acp-9-4797-2009, 2009.

Line 118, correct typo (Pommier et al., 2016).
Done

Line 141, actives to become active.
Changed.

Line 206, should read "Both biases are however" instead of "Both biases is howeve"
It reads now:
"The effects of both biases are, however, limited since most of HCOOH…"

Line 282, please specify which other studies.
This information is now available (in bold) in the following sentence:
"For the other regions, in addition to the values from Akagi et al. (2011), emission ratios were similarly calculated from emission factors given in other studies **(listed in Table 3**)."

Figure 2, include units in plots.

Figure 4, please include units in plots.
Figs 2 and 4 now include units, as hereafter:

[Figure]

**Figure 2: Time-series from 2008 to 2014 of the monthly means of IASI CO (blue) and HCOOH (red) total columns in $10^{18}$ molec/cm² and in $10^{16}$ molec/cm², respectively, FRP (black) in MegaWatts and the number of fires (magenta) from**

MODIS over the seven regions (AMA=Amazonia, AUS=Australia, IND = India, SEA = Southern East Asia, NAF= Northern Africa, SAF= Southern Africa, SIB= Siberia).

[Figure]

**Figure 4: Scatterplots between the IASI fire-affected HCOOH total columns (in $10^{16}$ molec/cm$^2$) and the CO total columns (in $10^{18}$ molec/cm$^2$) over the seven regions (AMA=Amazonia, AUS=Australia, IND = India, SEA = Southern East Asia, NAF= Northern Africa, SAF= Southern Africa, SIB= Siberia).The linear regression is represented by the blue line and the correlation coefficient is also provided for each region.**

---

## Author Comment (AC2) · 9 Aug 2017

[revised manuscript text omitted]

* Their "emission ratios" are requalified as enhancement ratios in this study since their ratios were not measured at the origin the fire emission but at high altitudes and/or further downwind of the fires.

**Table 3.** Enhancement ratio of HCOOH relative to CO (mol/mol) with its standard deviation and enhancement ratio of HCOOH relative to CO (mol/mol) by biome with its standard deviation calculated in this work. For each enhancement ratio by biome, the correlation coefficient and the number of MODIS hotspots are provided. The enhancement ratios are compared to emission ratios calculated from emission factors given in the literature for the seven studied regions. For the calculation of these emission ratios, the emission factors of CO for the corresponding fuel type given in Akagi et al. (2011) are used. Emission ratios of HCOOH relative to CO (mol/mol) calculated from the emission factors of HCOOH given in Akagi et al. (2011) for the corresponding fuel type are also provided.

| Region | Enhancement Ratio to CO (mol/mol) – this work | Enhancement Ratio to CO (mol/mol)[1] by biome[2] – this work | Emission Ratio to CO (mol/mol) calculated from $EF_{HCOOH}$ given in literature and using $EF_{CO}$ from Akagi et al. (2011) | Instrument used |
|---|---|---|---|---|
| AMA | $7.3\times10^{-3} \pm 0.08\times10^{-3}$ | $6.3\times10^{-3} \pm 0.22\times10^{-3}$ **(Evergreen Broadleaf forest, r=0.81, n = 454)** $3.0\times10^{-3} \pm 0.81\times10^{-3}$ **(Open shrubland, r=0.91, n = 5)** $7.0\times10^{-3} \pm 2.47\times10^{-3}$ **(Woody savanna, r=0.63, n = 14)** $7.6\times10^{-3} \pm 0.09\times10^{-3}$ **(Savanna, r=0.79, n = 3909)** $8.4\times10^{-3} \pm 0.39\times10^{-3}$ **(Grassland, r=0.88, n = 143)** $4.6\times10^{-3} \pm 0.35\times10^{-3}$ **(Cropland, r=0.88, n = 54)** | $1.8\times10^{-3}$ – Tropical forest (Yokelson et al., 2007 ; 2008)[3] $2.7\times10^{-3}$ – Savanna (Yokelson et al., 2007 ; 2008)[3] $2.0\times10^{-3}$ – Savanna (Akagi et al., 2011) $5.2\times10^{-3}$ – Tropical forest (Akagi et al., 2011) | Airborne FTIR (Yokelson et al., 2007) ; laboratory (Yokelson et al., 2008) catalogue |
| AUS | $11.1\times10^{-3} \pm 1.37\times10^{-3}$ | $5.7\times10^{-3} \pm 2.55\times10^{-3}$ **(Woody savanna, r=0.6, n = 11)** $11.2\times10^{-3} \pm 1.49\times10^{-3}$ **(Savanna, r=0.65, n = 80)** | $2.0\times10^{-3}$ – Savanna (Akagi et al., 2011) | catalogue |
| IND | $6.8\times10^{-3} \pm 0.44\times10^{-3}$ | $6.6\times10^{-3} \pm 0.77\times10^{-3}$ **(Woody savanna, r=0.65, n = 103)** $6.2\times10^{-3} \pm 0.62\times10^{-3}$ **(Cropland, r=0.58, n = 198)** $8.8\times10^{-3} \pm 1.19\times10^{-3}$ **(Cropland/Natural vegetation mosaic, r=0.85, n =23)** | $2.0\times10^{-3}$ – Savanna (Akagi et al., 2011) $2.7\times10^{-3}$ – Extratropical forest (Akagi et al., 2011) $6.0\times10^{-3}$ – Cropland (Akagi et al., 2011) | catalogue |
| SEA | $5.8\times10^{-3} \pm 0.15\times10^{-3}$ | $5.6\times10^{-3} \pm 0.20\times10^{-3}$ **(Evergreen Broadleaf forest, r=0.83, n = 334)** $6.3\times10^{-3} \pm 0.66\times10^{-3}$ **(Mixed forest, r=0.76, n = 70)** $6.2\times10^{-3} \pm 0.38\times10^{-3}$ **(Woody savanna, r=0.86, n = 99)** | $2.0\times10^{-3}$ – Savanna (Akagi et al., 2011) $2.7\times10^{-3}$ – Extratropical forest (Akagi et al., 2011) $6.0\times10^{-3}$ – Cropland (Akagi et al., 2011) | catalogue |

| | | | | |
|---|---|---|---|---|
| | | **7.1×10⁻³ ± 0.99×10⁻³ (Cropland/Natural vegetation mosaic, r=0.84, n =23)** | | |
| NAF | **4.0×10⁻³ ± 0.19×10⁻³** | **3.4×10⁻³ ± 0.63×10⁻³ (Evergreen Broadleaf forest, r=0.52, n = 78)**

**3.3×10⁻³ ± 0.28×10⁻³ (Woody savanna, r=0.44, n = 569)**

**4.4×10⁻³ ± 0.29×10⁻³ (Savanna, r=0.59, n = 441)**

**22.6×10⁻³ ± 11.06×10⁻³ (Cropland/Natural vegetation mosaic, r=0.67, n = 7)** | 2.0×10⁻³ – Savanna (Akagi et al., 2011) | catalogue |
| SAF | **5.0×10⁻³ ± 0.13×10⁻³** | **all hotspots are woody savanna** | 3.3×10⁻³ – Tropical forest (Sinha et al., 2004)[4]
4.8×10⁻³ – Savanna (Sinha et al., 2004)[4]

4.1×10⁻³ – Tropical forest (Yokelson et al., 2003)
6.0×10⁻³ – Savanna (Yokelson et al., 2003)

13×10⁻³ – Tropical forest (Rinsland et al., 2006)
19.2×10⁻³ – Savanna (Rinsland et al., 2006)

2.0×10⁻³ – Savanna (Akagi et al., 2011)
5.2×10⁻³ – Tropical forest (Akagi et al., 2011) | Airborne FTIR

Airborne FTIR

ACE-FTS

catalogue |
| SIB | **4.4×10⁻³ ± 0.09×10⁻³** | **4.0×10⁻³ ± 0.31×10⁻³ (Evergreen Needleaf forest, r=0.63, n = 245)**

**3.6×10⁻³ ± 0.16×10⁻³ (Deciduous Needleaf forest, r=0.66, n = 659)**

**3.4×10⁻³ ± 0.18×10⁻³ (Mixed forest, r=0.57, n = 759)**

**6.6×10⁻³ ± 0.48×10⁻³ (Open shrubland, r=0.76, n = 143)**

**6.0×10⁻³ ± 0.41×10⁻³ (Woody savanna, r=0.76, n = 155)**

**3.8×10⁻³ ± 0.65×10⁻³ (Permanent wetland, r=0.6, n = 63)** | 2.7×10⁻³ – Boreal forest (Akagi et al., 2011) | catalogue |

[1] Only the enhancement ratio to CO calculated from a scatterplot with a correlation coefficient higher than 0.4 are reported.
[2] The type of vegetation is defined by the land cover type data product (MCD12Q1).

[3] The EF$_{HCOOH}$ were corrected based on the comment from Yokelson et al. (2013) (EF$_{HCOOH}$ used: 0.281 for Yokelson et al. (2007); 0.2767 for Yokelson et al. (2008)).

[4] The mean of both EF$_{HCOOH}$ values provided in Sinha et al. (2004) were used for our EmR$_{HCOOH/CO}$ calculation

[Figure]

**Figure 1: Top: Number of MODIS fire hotspots with a confidence percentage higher or equal to 80%, averaged on a 0.5°×0.5° grid, for the period between 2008 and 2014. The blue boxes are the regions studied in this work. Middle: Classification of the land cover type from MODIS on the same grid and highlighting the studied regions in white. Each number corresponds to the type of vegetation. Only the data between 64°S and 84°N are available. Bottom: The IASI CO total column distribution (left) and the IASI HCOOH total column distribution (right), averaged between 2008 and 2014 and on the same grid.**

[Figure]

**Figure 2: Time-series from 2008 to 2014 of the monthly means of IASI CO (blue) and HCOOH (red) total columns in $10^{18}$ molec/cm$^2$ and in $10^{16}$ molec/cm$^2$, respectively, FRP (black) in MegaWatts and the number of fires (magenta) from MODIS over the seven regions (AMA=Amazonia, AUS=Australia, IND = India, SEA = Southern East Asia, NAF= Northern Africa, SAF= Southern Africa, SIB= Siberia).**

[Figure]

**Figure 3: Box and whisker plots showing mean (red central cross), median (red central line), and 25th and 75th percentile (blue box edges) of surface wind speed for each MODIS hotspot over the studied regions (AMA=Amazonia, AUS=Australia, IND = India, SEA = Southern East Asia, NAF= Northern Africa, SAF= Southern Africa, SIB= Siberia). The whiskers encompass values from 25th-1.5×(75th-25th) to the 75th+1.5× (75th-25th). This range of values corresponds to approximately 99.3% coverage if the data are normally distributed.**

[Figure]

**Figure 4: Scatterplots between the IASI fire-affected HCOOH total columns (in $10^{16}$ molec/cm²) and the CO total columns (in $10^{18}$ molec/cm²) over the seven regions (AMA=Amazonia, AUS=Australia, IND = India, SEA = Southern East Asia, NAF= Northern Africa, SAF= Southern Africa, SIB= Siberia). The linear regression is represented by the blue line and the correlation coefficient is also provided for each region.**

---

## Author Comment (AC3) · 9 Aug 2017

**Reviewer 2**

General Comments
This manuscript presents IASI measurements of formic acid between 2008 and 2014, and uses these data to determine enhancement ratios from biomass burning emissions over seven regions. HCOOH and CO total columns, MODIS fire counts, and ECMWF surface wind speeds are combined to identify enhancements due to biomass burning. Correlations between HCOOH and CO total columns are used to calculate the enhancement ratio in each region. These results suggest that production of HCOOH by Siberian forest fires may be underestimated by 60%, and provide some insights into sources and sinks of HCOOH in other regions studied.
The manuscript provides a useful contribution to the field, but is somewhat qualitative and speculative in places, as noted by the other reviewer. It also has many distracting grammatical errors and should be carefully reviewed and revised to correct these and to improve the clarity of the writing. I recommend publication in ACP after the comments below are addressed.

The authors would like to thank reviewer 2 for his careful reading of the manuscript and for his thorough review. A detailed point by point reply (in blue) is provided hereafter.
As suggested by the first reviewer, an additional work has been done by using backward trajectories from HYSPLIT and land cover information from MODIS.
Moreover the text has been revised. Thus, in addition to our answers, we suggest that the reviewer reads the revised manuscript since Sect. 5.1 and 5.2 were largely rewritten.

A lot of MODIS hotspots have been studied in this paper, in total it represents 9628 hotspots. It is difficult to calculate backward trajectories for each hotspot, especially as different altitude ranges need to be tested since the vertical sensitivity of IASI (CO & HCOOH) is located in the free troposphere.
However, illustrative tests were done to show the distinct origins of the air masses at different locations, periods of the year and altitudes of the plume. Specifically, 5 hotspots have been chosen randomly for each region and 3 different altitudes have been selected: 500 m (thus close to the surface), 2000 m and 5000 m (representing the free troposphere). In total, this represents 105 trajectories.
These trajectories show that the air masses initialized at 500 and 2000m are mainly influencing by air masses close to the surface, confirming an origin near the source of our IASI fire-affected columns. It also shows the difficulty to estimate the origin of the air masses without an accurate knowledge of the altitude of the plumes.
These trajectories were plotted through the HYSPLIT online service:

[Figure]

Fig 1. 5-day backward trajectories from HYPSLIT online service calculated at 3 altitudes: 500 m (red), 2000 m (blue) and 5000 m (green), for 5 hotspots chosen randomly over the 7 regions studied in the paper. The parameters characterizing each MODIS hotpots are summarized in the following table. The meteorological fields are from GDAS at 1°×1° horizontal resolution.

[Figure]

Fig 1. Continue

**Tab.** Characteristic of each MODIS hotspot used for the trajectories plotted in the previous figure. The dates, the time recorded by the instrument and the coordinates for each hotspot are written.

| AMA |
|-----|
| 20100906 / hour (UTC)=13 / lat=-6.476 - lon=-49.71 |
| 20120220 / hour (UTC)=2 / lat=-9.937 - lon=-59.911 |
| 20120817 / hour (UTC)=13 / lat=-5.839 - lon=-46.987 |
| 20130921 / hour (UTC)=13 / lat=-11.777 - lon=-50.871 |
| 20131011 / hour (UTC)=13 / lat=-11.09 - lon=-48.229 |
| **AUS** |
| 20130910 / hour (UTC)=0 / lat=-12.841 - lon=132.327 |
| 20130913 / hour (UTC)=1 / lat=-12.916 - lon=132.323 |
| 20130927 / hour (UTC)=1 / lat=-12.215 - lon=131.169 |
| 20131002 / hour (UTC)=1 / lat=-13.696 - lon=131.59 |
| 20131003 / hour (UTC)=0 / lat=-13.428 - lon=133.844 |
| **IND** |
| 20090407 / hour (UTC)=4 / lat=21.503 - lon=82.645 |
| 20090407 / hour (UTC)=4 / lat=20.067 - lon=84.175 |
| 20110317 / hour (UTC)=5 / lat=21.572 - lon=77.328 |
| 20120416 / hour (UTC)=5 / lat=22.166 - lon=77.749 |
| 20140420 / hour (UTC)=5 / lat=23.17 - lon=75.544 |
| **SEA** |
| 20110414 / hour (UTC)=3 / lat=22.681 - lon=96.801 |
| 20120402 / hour (UTC)=3 / lat=19.435 - lon=101.908 |
| 20130315 / hour (UTC)=3 / lat=21.594 - lon=100.047 |
| 20130329 / hour (UTC)=3 / lat=21.426 - lon=98.744 |
| 20140314 / hour (UTC)=3 / lat=19.8 - lon=100.375 |

| NAF |
|---|
| 20080108  / hour (UTC)=8  / lat=5.672 - lon=28.741 |
| 20100116  / hour (UTC)=8  / lat=8.419 - lon=19.196 |
| 20110222  / hour (UTC)=8  / lat=6.148 - lon=29.351 |
| 20121205  / hour (UTC)=8  / lat=9.168 - lon=19.859 |
| 20140203  / hour (UTC)=8  / lat=6.024 - lon=17.002 |
| **SAF** |
| 20090622  / hour (UTC)=8  / lat=-7.339 - lon=21.495 |
| 20100815  / hour (UTC)=9  / lat=-5.351 - lon=15.48 |
| 20111004  / hour (UTC)=20  / lat=-5.129 - lon=27.708 |
| 20120718  / hour (UTC)=20  / lat=-8.822 - lon=16.974 |
| 20140805  / hour (UTC)=8  / lat=-8.721 - lon=18.288 |
| **SIB** |
| 20110602  / hour (UTC)=4  / lat=59.259 - lon=98.286 |
| 20110609  / hour (UTC)=4  / lat=57.882 - lon=99.823 |
| 20120623  / hour (UTC)=6  / lat=62.39 - lon=85.655 |
| 20130807  / hour (UTC)=3  / lat=64.98 - lon=118.109 |
| 20130819  / hour (UTC)=4  / lat=55.39 - lon=106.945 |

We have added the sentences (in bold) in our Section 5.1.2:
**"A few backward trajectories (along 5 days, not shown) have been calculated for our hotspots with the online version of the HYSPLIT atmospheric transport and dispersion modeling system (Rolph, 2017). These trajectories, initialized at different altitudes, confirm a main origin close to the surface of our IASI fire-affected columns. It is however impossible to properly compare the origin of the air masses with previous studies as our studied period (2008-2014) or our studied fires do not necessarily match with plumes described in other publications. It is also difficult to estimate the age of our studied air masses by gathering the plumes during a 7-yr period and without an accurate knowledge of the altitude of the plumes."**

And:

"One possible explanation is the multi-origin of the plumes studied by Rinsland et al. (2006), since, based on their backward trajectories, their plumes could be influenced by biomass burning originating from Southern Africa and/or from Southern America. **The travel during the few days across the Atlantic Ocean may explain the change in their $ER_{(HCOOH/CO)}$."**

With the corresponding reference:
Rolph, G.D.: Real-time Environmental Applications and Display sYstem (READY) Website (http://www.ready.noaa.gov). NOAA Air Resources Laboratory, College Park, MD, 2017.

We also have discovered that the information about the type of vegetation was available in the MODIS products.
In "Section 3. MODIS" we have added this paragraph:
"To characterize each MODIS hotspot by the type of fuel burned, the Global Mosaics of the standard MODIS land cover type data product (MCD12Q1) in the IGBP Land Cover Type Classification (Friedl et al., 2010; Channan et al., 2014) with a $0.5° \times 0.5°$ horizontal resolution has also been used (http://glcf.umd.edu/data/lc/). As the annual variability in this product is limited (not shown) and since the period available (from 2001 to 2012) does not fully match

the period of the IASI mission, only the data for 2012 have been used. Whitburn et al. (2017) have also used this MCD12Q1 product to determine their IASI-derived $NH_3$ enhancement ratios by vegetation types."

With the corresponding references:

Channan, S., Collins, K., and Emanuel, W. R.,Global mosaics of the standard MODIS land cover type data. University of Maryland and the Pacific Northwest National Laboratory, College Park, Maryland, USA, 2014.
And
Friedl, M. A., Sulla-Menashe, D., Tan, B., Schneider, A., Ramankutty, N., Sibley, A. and Huang, X., MODIS Collection 5 global land cover: Algorithm refinements and characterization of new datasets, 2001-2012, Collection 5.1 IGBP Land Cover,  Remote Sensing of Environment, 114 , 168–182, doi:10.1016/j.rse.2009.08.016, 2010.
And
Whitburn, S., Van Damme, M., Clarisse, L., Hurtmans, D., Clerbaux, C., and Coheur, P.-F.: IASI-derived NH3 enhancement ratios relative to CO for the tropical biomass burning regions, Atmos. Chem. Phys. Discuss., https://doi.org/10.5194/acp-2017-331, in review, 2017.

We have also added this sentence in Section 4.2:
"The classification of the vegetation from the MODIS product has also been used for a detailed analysis of the enhancement ratios for these regions (Fig. 1)."
And Fig. 1 has been modified as below:

[Figure]

**Figure 1: Top: Number of MODIS fire hotspots with a confidence percentage higher or equal to 80%, averaged on a 0.5°×0.5° grid, for the period between 2008 and 2014. The blue boxes are the regions studied in this work. Middle: Classification of the land cover type from MODIS on the same grid and highlighting the studied regions in white. Each number corresponds to the type of vegetation. Only the data between 64°S and 84°N are available. Bottom: The IASI CO total column distribution (left) and the IASI HCOOH total column distribution (right), averaged between 2008 and 2014 and on the same grid.**

Section 5.2 was also rewritten and now named "5.2. Analysis based on the type of vegetation" since ER (HCOOH/CO) by biome have also been added in Table 3 as below:

**Table 3.** Enhancement ratio of HCOOH relative to CO (mol/mol) with its standard deviation and enhancement ratio of HCOOH relative to CO (mol/mol) by biome with its standard deviation calculated in this work. For each enhancement ratio by biome, the correlation coefficient and the number of MODIS hotspots are provided. The enhancement ratios are compared to emission ratios calculated from emission factors given in the literature for the seven studied regions. For the calculation of these emission ratios, the emission factors of CO for the corresponding fuel type given in Akagi et al. (2011) are used. Emission ratios of HCOOH relative to CO (mol/mol) calculated from the emission factors of HCOOH given in Akagi et al. (2011) for the corresponding fuel type are also provided.

| Region | Enhancement Ratio to CO (mol/mol) – this work | Enhancement Ratio to CO (mol/mol)[1] by biome[2] – this work | Emission Ratio to CO (mol/mol) calculated from $EF_{HCOOH}$ given in literature and using $EF_{CO}$ from Akagi et al. (2011) | Instrument used |
|---|---|---|---|---|
| AMA | $7.3 \times 10^{-3} \pm 0.08 \times 10^{-3}$ | $6.3 \times 10^{-3} \pm 0.22 \times 10^{-3}$ **(Evergreen Broadleaf forest, r=0.81, n = 454)**

$3.0 \times 10^{-3} \pm 0.81 \times 10^{-3}$ **(Open shrubland, r=0.91, n = 5)**

$7.0 \times 10^{-3} \pm 2.47 \times 10^{-3}$ **(Woody savanna, r=0.63, n = 14)**

$7.6 \times 10^{-3} \pm 0.09 \times 10^{-3}$ **(Savanna, r=0.79, n = 3909)**

$8.4 \times 10^{-3} \pm 0.39 \times 10^{-3}$ **(Grassland, r=0.88, n = 143)**

$4.6 \times 10^{-3} \pm 0.35 \times 10^{-3}$ **(Cropland, r=0.88, n = 54)** | $1.8 \times 10^{-3}$ – Tropical forest (Yokelson et al., 2007 ; 2008)[3]
$2.7 \times 10^{-3}$ – Savanna (Yokelson et al., 2007 ; 2008)[3]

$2.0 \times 10^{-3}$ – Savanna (Akagi et al., 2011)
$5.2 \times 10^{-3}$ – Tropical forest (Akagi et al., 2011) | Airborne FTIR (Yokelson et al., 2007) ; laboratory (Yokelson et al., 2008)

catalogue |
| AUS | $11.1 \times 10^{-3} \pm 1.37 \times 10^{-3}$ | $5.7 \times 10^{-3} \pm 2.55 \times 10^{-3}$ **(Woody savanna, r=0.6, n = 11)**

$11.2 \times 10^{-3} \pm 1.49 \times 10^{-3}$ **(Savanna, r=0.65, n = 80)** | $2.0 \times 10^{-3}$ – Savanna (Akagi et al., 2011) | catalogue |
| IND | $6.8 \times 10^{-3} \pm 0.44 \times 10^{-3}$ | $6.6 \times 10^{-3} \pm 0.77 \times 10^{-3}$ **(Woody savanna, r=0.65, n = 103)**

$6.2 \times 10^{-3} \pm 0.62 \times 10^{-3}$ **(Cropland, r=0.58, n = 198)**

$8.8 \times 10^{-3} \pm 1.19 \times 10^{-3}$ **(Cropland/Natural vegetation mosaic, r=0.85, n =23)** | $2.0 \times 10^{-3}$ – Savanna (Akagi et al., 2011)
$2.7 \times 10^{-3}$ – Extratropical forest (Akagi et al., 2011)
$6.0 \times 10^{-3}$ – Cropland (Akagi et al., 2011) | catalogue |
| SEA | $5.8 \times 10^{-3} \pm 0.15 \times 10^{-3}$ | $5.6 \times 10^{-3} \pm 0.20 \times 10^{-3}$ **(Evergreen Broadleaf forest, r=0.83, n = 334)** | $2.0 \times 10^{-3}$ – Savanna (Akagi et al., 2011)
$2.7 \times 10^{-3}$ – Extratropical forest (Akagi et al., 2011)
$6.0 \times 10^{-3}$ – Cropland (Akagi et al., 2011) | catalogue |

| | | | | |
|---|---|---|---|---|
| | | $6.3\times10^{-3} \pm 0.66\times10^{-3}$ (Mixed forest, r=0.76, n = 70)

$6.2\times10^{-3} \pm 0.38\times10^{-3}$ (Woody savanna, r=0.86, n = 99)

$7.1\times10^{-3} \pm 0.99\times10^{-3}$ (Cropland/Natural vegetation mosaic, r=0.84, n =23) | | |
| NAF | $4.0\times10^{-3} \pm 0.19\times10^{-3}$ | $3.4\times10^{-3} \pm 0.63\times10^{-3}$ (Evergreen Broadleaf forest, r=0.52, n = 78)

$3.3\times10^{-3} \pm 0.28\times10^{-3}$ (Woody savanna, r=0.44, n = 569)

$4.4\times10^{-3} \pm 0.29\times10^{-3}$ (Savanna, r=0.59, n = 441)

$22.6\times10^{-3} \pm 11.06\times10^{-3}$ (Cropland/Natural vegetation mosaic, r=0.67, n = 7) | $2.0\times10^{-3}$ – Savanna (Akagi et al., 2011) | catalogue |
| SAF | $5.0\times10^{-3} \pm 0.13\times10^{-3}$ | all hotspots are woody savanna | $3.3\times10^{-3}$ – Tropical forest (Sinha et al., 2004)[4]
$4.8\times10^{-3}$ – Savanna (Sinha et al., 2004)[4] | Airborne FTIR |
| | | | $4.1\times10^{-3}$ – Tropical forest (Yokelson et al., 2003)
$6.0\times10^{-3}$ – Savanna (Yokelson et al., 2003) | Airborne FTIR |
| | | | $13\times10^{-3}$ – Tropical forest (Rinsland et al., 2006)
$19.2\times10^{-3}$ – Savanna (Rinsland et al., 2006) | ACE-FTS |
| | | | $2.0\times10^{-3}$ – Savanna (Akagi et al., 2011)
$5.2\times10^{-3}$ – Tropical forest (Akagi et al., 2011) | catalogue |
| SIB | $4.4\times10^{-3} \pm 0.09\times10^{-3}$ | $4.0\times10^{-3} \pm 0.31\times10^{-3}$ (Evergreen Needleaf forest, r=0.63, n = 245)

$3.6\times10^{-3} \pm 0.16\times10^{-3}$ (Deciduous Needleaf forest, r=0.66, n = 659)

$3.4\times10^{-3} \pm 0.18\times10^{-3}$ (Mixed forest, r=0.57, n = 759) | $2.7\times10^{-3}$ – Boreal forest (Akagi et al., 2011) | catalogue |

$6.6\times10^{-3} \pm 0.48\times10^{-3}$
**(Open shrubland,**
**r=0.76, n = 143)**

$6.0\times10^{-3} \pm 0.41\times10^{-3}$
**(Woody savanna,**
**r=0.76, n = 155)**

$3.8\times10^{-3} \pm 0.65\times10^{-3}$
**(Permanent wetland,**
**r=0.6, n = 63)**

[1] Only the enhancement ratio to CO calculated from a scatterplot with a correlation coefficient higher than 0.4 are reported.

[2] The type of vegetation is defined by the land cover type data product (MCD12Q1).

[3] The $EF_{HCOOH}$ were corrected based on the comment from Yokelson et al. (2013) ($EF_{HCOOH}$ used: 0.281 for Yokelson et al. (2007); 0.2767 for Yokelson et al. (2008)).

[4] The mean of both $EF_{HCOOH}$ values provided in Sinha et al. (2004) were used for our $EmR_{HCOOH/CO}$ calculation

We have added these sentences at the end of the Section 5.2:
"In addition to the $EmR_{(HCOOH/CO)}$ calculated from the $EF_{HCOOH}$ given in the literature, a classification for our $ER_{(HCOOH/CO)}$ has also been done, based on the data from the MCD12Q1 product. As each hotspot is associated with a land cover value defined by the MCD12Q1 product, enhancement ratios by biome have been calculated. The limitations of this dataset are its coarse resolution (0.5° × 0.5°) and the lack of seasonal variation. It gives however a supplementary information on the type of fuel burned identified by MODIS. The corresponding $ER_{(HCOOH/CO)}$ are provided in Table 3. Only the values calculated from a scatterplot with a correlation coefficient higher than 0.4 are reported."

And

"In general, the $ER_{(HCOOH/CO)}$ calculated for a specific biome varies with the regions. This shows that the type of vegetation is not the only factor influencing the $ER_{(HCOOH/CO)}$. The ongoing chemistry within a plume is important and the age of the air masses impact the level of HCOOH and CO in the plumes."

We have also added these sentences in the abstract:
"An additional classification of the enhancement ratios by type of fuel burned is also provided, showing a diverse origin of the plumes sampled by IASI, especially over Amazonia and Siberia. The variability in the enhancement ratios by biome over the different regions show that the levels of HCOOH and CO do not only depend on the fuel types."
And in the conclusion:
"Finally, the estimation of the $ER_{(HCOOH/CO)}$ calculated by the type of vegetation burned, as referenced in the MODIS product, varies with the regions. This shows that other parameters than the type of fuel burned also influence the $ER_{(HCOOH/CO)}$."

Specific Comments
Page 1, line 1 – The title is awkwardly phrased. Why just a "Possibility" for IASI to detect HCOOH in biomass burning plumes? "document" should be replaced by "measure" or "detect". A better title might be something like: "Detection of HCOOH from biomass burning plumes by the Infrared Atmospheric Sounding Interferometer"
The title is now:
"Determination of enhancement ratios of HCOOH relative to CO in biomass burning plumes by the Infrared Atmospheric Sounding Interferometer (IASI)."

Page 1, lines 25-27 – Make clear whether this underestimation for Siberian forest fires is in the IASI HCOOH or other studies or both. This seems rather speculative based on the results presented in the paper.

This information has been deleted from the abstract but we have added these sentences (in bold) in the conclusions:

"The underestimation by 60% over Siberia is consistent with conclusions given in R'Honi et al. (2013). **The calculation of the ER$_{(HCOOH/CO)}$ by biome shows that Siberian plumes are related to the burning of six different vegetation classes. The underestimation reported is thus difficult to confirm without the use of a chemical transport model."**

We have also written in Section 5.2:
"These hypotheses in biased emissions and/or secondary production need, however, to be verified with modeling studies."

Page 1, lines 27-29 – Rewrite this last sentence for clarity.

Done. Now it reads:
"In comparison **with referenced emission ratios**, it is also shown that the selected agricultural burning plumes captured by IASI over India and Southern East Asia correspond to recent plumes where the chemistry or the sink does not occur."

Page 5, line 185 – Why is 1.44 m/s used as a threshold?

As explained in the following sentence: "This value of 1.44 m/s for the surface wind speed corresponds to the 25th percentile of the distribution of the three regions characterized by the lowest surface wind speed (Fig. 3)."

Page 6, lines 210-212 – Please clarify this discussion. It is not clear how a better detection limit "minimizes the bias with the lowest columns", nor what suggests "a negligible effect of the low column biases".

We agreed it was confusing. Now the sentences are (the modifications are highlighted in bold):

[revised manuscript text omitted]

Line 239 – Where were the plumes sampled by Yokelson et al.?
The plumes were over Zambia, Zimbabwe and South Africa. This information is now included in the sentence (in bold):
"The $ER_{(HCOOH/CO)}$ from our work is also 15% lower than the $EmR_{(HCOOH/CO)}$ in Yokelson et al. (2003) ($5.9 \times 10^{-3} \pm 2.2 \times 10^{-3}$  mol/mol) **who calculated their value within plumes over Zambia, Zimbabwe and South Africa**."

Page 7, lines 243-244 – What was the approach developed by Chaliyakunnel et al. (2016) to determine pyrogenic ER(HCOOH/CO)? It is not clear what is meant "by reducing the impact of the mix with the ambient air".

An explanation of their approach is now added (in bold):

"**To do so, they calculated the $ER_{(HCOOH/CO)}$ in the vicinity of fire count from MODIS (averaged in a cell having the resolution of the GEOS-Chem model, i.e. $2° \times 2.5°$) and they differentiated this value with a background $ER_{(HCOOH/CO)}$ defined by the concentrations distant from these fires.** They concluded that their most reliable value on the amount of HCOOH produced from fire emissions was obtained for African fires."

Page 7, lines 269-271 – Revise this poorly written paragraph. It is not clear what is meant by either sentence.

As explained as introduction of our answers, section 5.2 was rewritten and the title of section has been changed.

To answer the question about lines 269-271, the new lines are (the changes are in bold):

**"5.2. Analysis based on the type of vegetation**

We have complemented our comparison of the enhancement ratios **by comparing our ratios to emissions ratios calculated** from emission factors found in literature. The main argument to perform such comparison is the lack of measurements of enhancement ratios over IND and SEA. **Furthermore, such comparison from emission factors facilitates an analysis based on hypothesis about the type of vegetation burned."**

Page 7, lines 275-279 – Why can't the decay be taken into account by considering the exponential decrease between emission and detection using relative lifetimes, e.g., Viatte et al. (2015) and references therein?

Each MODIS hotspot is characterized by a mean CO total column and a mean HCOOH total column. These averages are calculated along 5 hours. During 5 hours, the chemistry may already occur and it is the reason why we have written that the decay of these compounds could not be taken into account in our methodology.

Moreover, without to know the accurate altitude of the plumes, it is challenging to calculate the age of the air masses.

Sections 5.1 and 5.2 – Both sections discuss enhancement ratios and emission ratios, including comparisons with other studies, e.g., on page 8, there is additional discussion of ER although the title suggests that Section 5.2 is about EmR. These sections could be more clearly differentiated.

Both sections have been changed.

Now there are:
5. Analysis of the data over the fire regions
5.1. Determination of the enhancement ratios
5.1.1 General analysis
5.1.2 Analysis over each region
5.2. Analysis based on the type of vegetation

Page 9, lines 358-359 – Arguably, such an intercomparison could have been included in this study.

An inter-comparison has to be done but it is beyond the scope of this paper. It will be a subject for a next study. It is the reason why it was mentioned in the conclusion.

Technical Corrections
Page 1, line 19 – add comma after "(MODIS)"
Done

Page 1, line 26 – add comma after "forest fires"
The sentence has been deleted.

Page 1, line 34 – delete "for"
Done

Page 2, line 46 – Rewrite this sentence. Not clear what is meant by "as on the oxidizing power..."
A complementary information is now provided. The changes are shown in bold:
"… on the oxidizing **capacity** of the atmosphere **(i.e. the chemistry of OH in cloud water - Jacob, 1986; the heterogeneous oxidation of organic aerosols - Paulot et al., 2011**)"

and we have added this reference:
Jacob, D.: Chemistry of OH in remote clouds and its role in the production of formic acid and peroxymonosulfate, J. Geophys. Res., 91, 9807–9826, 1986.

Page 2, line 55 – "hence depend on"
Corrected

Page 2, line 67 – change "as with" to "including" or "such as"
"Including" is now used.

Page 2, line 69 – delete "with the"
Done

Page 2, line 70 – "Atmospheric Chemistry Experiment – Fourier Transform Spectrometer (ACE-FTS)"
Changed

Page 2, line 72 – I think this means "(MIPAS) limb instrument, which is sensitive to altitudes down to ~ 10 km" (rather than only sensitive at 10 km)
Grutter et al. (2010) – the cited reference – shows distributions and time-series at 10 km. Most of their profiles start at 8 km, and thus we kept the sentence:
"… Michelson Interferometer for Passive Atmospheric Sounding (MIPAS) limb instrument which is sensitive to around 10 km (Grutter et al., 2010)."

Page 2, line 74 – "compared to ground-based and airborne"
Corrected.

Page 2, line 75 – "allows observation of remote regions"
"which allows observing remote regions" is now changed by "allows observation of remote regions".

Page 2, line 77 – "ratios of HCOOH relative to CO over"
Changed.

This change has also done in the title of the tables.

Page 3, lines 93-94 – add space before K, as done for other units like km, cm-1, etc.
Done

Page 3, line 97 – Isn't the lifetime of CO closer to two months than several weeks?
The lifetime depends on the season and on the location. We clarified this point by changing "several weeks" by "a few weeks to a few months depending on latitude and time of year."

Page 3, line 113 – "in more detail"
Corrected

Page 3, lines 117-118 – "which is less than 35% for total columns smaller than..."
Changed

Page 4, line 123 – "hotspots"
"s" has been added.

Page 4, line 123 – MODIS has already been defined
That is correct. Thank you for noticing it.

Page 4, line 129 – "which, for each detected fire pixel, includes the ..."
Changed.

Page 4, line 132 – Last sentence doesn't need to be a separate paragraph.
Changed.

Page 4, line 141 – "most active in terms of actual fires but are still of interest. The first ..."
These four sentences about importance of biomass burning in India and Siberia could also be rewritten for clarity.
The sentence has been changed as:
"Among these regions, India and Siberia do not represent the most active regions in terms of number of fires. It seemed however important to also investigate them."

Page 4, line 144 – "over some years, such as during summer 2010"
Changed

Page 4, line 154 – "(correlation coefficient, r, from"
Changed

Page 4, line 155 – "the impact of sources other than biomass burning"
Changed.

Page 4, line 156 – "also have"
Changed

Page 4, line 160 – "The large region selected over Siberia"
Changed

Page 4, line 161 – "other regions, such as polluted"

Changed

Page 5, line 170 – add comma after "criteria"
Done

Page 5, line 171 – "in Table 1. The smaller correlation coefficients, i.e., less than 0.7, are found"
Changed

Page 5, line 172 – "the HCOOH and CO columns"
Changed

Page 5, line 178 – "assign" rather than "attribute" ?
Changed

Page 5, line 179 – ECMWF has already been defined
It was not defined previously, except in the abstract. Thus we have decided to keep the definition in this line.

Page 5, line 182 – "(r close to 0.8)"
"r" has been added.

Page 5, line 183 – Clarify that the low mean and median refer to surface wind speed. Also rewrite the sentence on line 184 for clarity.
The sentence has been changed as:
"IND has also a low mean and median **surface wind speed** but the distribution of **this** surface wind speed **over IND** is more spread out than for **AMA, SEA and SAF**."

Page 5, line 186 and elsewhere through the manscript– "in Table 2" ? Does ACP accept Tab. as an abbreviation for Table?
"Tab" has been changed by "Table" everywhere through the manuscript.

Page 5, line 197 – "than using only the columns"
Changed

Page 5, line 198 – "for each measurement pair"
Corrected

Page 6, line 201 – "so comparison with previous work is ... over another"
Corrected.

Page 6, line 203 – should globally be generally?
"globally" was changed by "generally".

Page 6, line 206 – "The effects of both biases are, however, limited"
The sentence has been changed as requested.

Page 6, line 211 – "an improved [or a lower?] detection limit"
That's correct; in this case, improved means lower. To clarify it, we have changed "improve" by "reduce". Now it reads:

"Indeed, the increase in the thermal contrast (i.e. the temperature difference between the surface and the first layer in the retrieved profile) leads to **reducing** the detection limit as shown in Pommier al. (2016)."

Page 6, line 222 – "same plume as"
Corrected

Page 6, line 231 – trajectories
Corrected

Page 6, line 235 – "reasons for the agreement"
The sentence has been changed:
"However, the difference in geometry cannot explain why we find an agreement with the ACE-FTS measurements values reported by Coheur et al. (2007) and a disagreement with those from Rinsland et al. (2006)."

Page 6, lines 241-242 – "Conversely, the … from IASI is twice that of Chaliyakunnel"
Changed. The sentence is now:
"**Conversely,** the $ER_{(HCOOH/CO)}$ retrieved **from IASI is twice that of** Chaliyakunnel et al. (2016) $(2.6 \times 10^{-3} \pm 0.3 \times 10^{-3}$ mol/mol)."

Page 7, line 247 – No need for a new paragraph here.
We have preferred to keep this new paragraph since it corresponds to the analysis of the results over NAF and the previous paragraphs are about SAF.

Page 7, line 248 – "worth noting"
"Reminding" is replaced by "noting" as requested.

Page 7, line 251 – "and that of Paton-Walsh (2005) may be explained"
Changed. The sentence is now: "The difference between our work a**nd that of Paton-Walsh (2005) may be explained** by the different origin of the probed plume."

Page 7, line 254 – quantify "quite uncertain"
The value is now given in text:
"… a quite uncertain value is reported **$(4.5 \times 10^{-3} \pm 5.1 \times 10^{-3}$ mol/mol)**,…"

Page 7, line 280 – "For both the IND"
The comma has been deleted as requested.

Page 8, line 287 – Equation
Changed

Page 8, line 289 – "composed of tropical"
Page 8, line 292 – "composed of cropland"
Both were corrected.

Page 8, line 293 – "characterized by an"
Corrected.

Page 8, line 300 – "(2004) both used the same"

Changed.

Page 8, line 307 – "twice the value" [also specify whether ER or EmR from Akagi]
We agreed that this sentence was confusing. We changed "value" by "EmR". Now it reads:
"Over Northern Africa, our $ER_{(HCOOH/CO)}$ **is twice as large as** the $\mathbf{EmR_{(HCOOH/CO)}}$ provided by Akagi et al. (2011), probably due to the lower correlation found in our scatterplot."

Page 8, line 308 – "It is highly"
Changed.

Page 8, line 314 – "forest fire plumes"
Changed.

Page 9, line 336 – "difficulties ... are"
Corrected

Page 9, line 338 – "using satellite, airborne, or FTIR measurements"
Changed

Page 9, line 346 – "A very good agreement was found" in what? Specify.
The information has been added (in bold):
"A very good agreement **in $ER_{(HCOOH/CO)}$** was found over Amazonia, especially with the work done by Chaliyakunnel et al. (2016) who determinated pyrogenic $ER_{(HCOOH/CO)}$."

Page 9, line 349 – Replace "delicate" with a better description.
We replaced "delicate" by "complicated":
"The analysis over Australia is however **complicated** as our $ER_{(HCOOH/CO)}$ approximately corresponds…"

Page 9, line 355 – "a modelling study could be"
"work" has been replaced by "study".

Page 9, line 357 – times
"s" has been added.

Page 9, line 358 – "instruments such as"
"Such" has been added.

Page 10, line 367 – Isn't IASI an instrument, not a mission?
There are 3 similar instruments. So IASI is both an instrument and a long term mission.

Page 10, line 372 – "for free access"
Corrected

Page 10, lines 385 and 387 – Inconsistent formatting of references for the same journal.
Thanks for this observation. The doi was missing for the first reference. Now the references are:
Andrews, D. U., Heazlewood, B. R., Maccarone, A. T., Conroy, T., Payne, R. J., Jordan, M. J. T., and Kable, S. H.: Photo-tautomerization of acetaldehyde to vinyl alcohol: a potential route to tropospheric acids, Science, 337, 1203–1206, doi:10.1126/science.1220712, 2012.

Beirle, S., K. F. Boersma, U. Platt, M. G. Lawrence, and T. Wagner : Megacity emissions and lifetimes of nitrogen oxides probed from space, Science, 333, 1737–1739, doi:10.1126/science.1207824, 2011.

Page 17, caption line 2 – "over the 7 [seven] regions studied. IASI data are"
7 has been replaced by "seven".

Page 17 – Tab. or Table ?
As for your previous comment, "Tab" has been changed by "Table" everywhere through the manuscript.

Pages 18 and 19, table headings – "HCOOH/CO Enhancement/Emission Ratio ..." would be a better title
The table headings are:

**Table 2.** Enhancement ratio of HCOOH relative to CO (mol/mol) with its standard deviation compared to enhancement ratios of HCOOH relative to CO and emissions ratios of HCOOH reported in the literature for the seven studied regions.

**Table 3.** Enhancement ratio of HCOOH relative to CO (mol/mol) with its standard deviation and enhancement ratio of HCOOH relative to CO (mol/mol) by biome with its standard deviation calculated in this work. For each enhancement ratio by biome, the correlation coefficient and the number of MODIS hotspots are provided. The enhancement ratios are compared to emission ratios calculated from emission factors given in the literature for the seven studied regions. For the calculation of these emission ratios, the emission factors of CO for the corresponding fuel type given in Akagi et al. (2011) are used. Emission ratios of HCOOH relative to CO (mol/mol) calculated from the emission factors of HCOOH given in Akagi et al. (2011) for the corresponding fuel type are also provided.

Page 18, Table 3 – Left justify all the table entries
Done

Page 19, caption line 2 – "in the literature". Also, rewrite the full caption for conciseness and clarity, e.g., HCOOH/CO enhancement ratio, etc.
See our answer about the table headings.

Page 20, caption line 3 – "column distribution ... column distribution"
Corrected

Page 21, Figure 2 and page 22, Figure 4– Preferable to have units on the y-axis labels, rather than just in the caption.
Figs 2 and 4 now include units, as hereafter:

[Figure]

**Figure 2: Time-series from 2008 to 2014 of the monthly means of IASI CO (blue) and HCOOH (red) total columns in $10^{18}$ molec/cm$^2$ and in $10^{16}$ molec/cm$^2$, respectively, FRP (black) in MegaWatts and the number of fires (magenta) from MODIS over the seven regions (AMA=Amazonia, AUS=Australia, IND = India, SEA = Southern East Asia, NAF= Northern Africa, SAF= Southern Africa, SIB= Siberia).**

[Figure]

**Figure 4: Scatterplots between the IASI fire-affected HCOOH total columns (in $10^{16}$ molec/cm$^2$) and the CO total columns (in $10^{18}$ molec/cm$^2$) over the seven regions (AMA=Amazonia, AUS=Australia, IND = India, SEA = Southern East Asia, NAF= Northern Africa, SAF= Southern Africa, SIB= Siberia).The linear regression is represented by the blue line and the correlation coefficient is also provided for each region.**

Page 22, Figure 3, caption line 4 – Clarify text describing the percentiles

The sentences "The whiskers encompass values from 25th-1.5×(75th-25th) to the 75th+1.5× (75th-25th). This range covers more than 99% of a normally distributed data set." Have been changed by:

"The whiskers encompass values from 25[th]-1.5×(75[th]-25[th]) to the 75[th]+1.5× (75[th]-25[th]). This range of values corresponds to approximately 99.3% coverage if the data are normally distributed. "

---

## Author Comment (AC4) · 9 Aug 2017

The comment was uploaded in the form of a supplement:
https://www.atmos-chem-phys-discuss.net/acp-2017-126/acp-2017-126-AC4-supplement.pdf